# Dislocation exhaustion and ultra-hardening of nanograined metals by phase transformation at grain boundaries

Shangshu Wu[1,6], Zongde Kou[1,6], Qingquan Lai [1,5] ✉, Si Lan [1],
Shyam Swaroop Katnagallu[2], Horst Hahn [1,2], Shabnam Taheriniya[3],
Gerhard Wilde [1,3], Herbert Gleiter[1,3,4] & Tao Feng [1] ✉

The development of high-strength metals has driven the endeavor of pushing the limit of grain size (d) reduction according to the Hall-Petch law. But the continuous grain refinement is particularly challenging, raising also the problem of inverse Hall-Petch effect. Here, we show that the nanograined metals (NMs) with d of tens of nanometers could be strengthened to the level comparable to or even beyond that of the extremely-fine NMs (d ~ 5 nm) attributing to the dislocation exhaustion. We design the Fe-Ni NM with intergranular Ni enrichment. The results show triggering of structural transformation at grain boundaries (GBs) at low temperature, which consumes lattice dislocations significantly. Therefore, the plasticity in the dislocation-exhausted NMs is suggested to be dominated by the activation of GB dislocation sources, leading to the ultra-hardening effect. This approach demonstrates a new pathway to explore NMs with desired properties by tailoring phase transformations via GB physico-chemical engineering.

Grain refinement is one of the prevailing strengthening methods as depicted by the Hall–Petch relationship[1,2]: the strength is inversely proportional to the square root of grain size (d). The pursuit of high-strength metals has led to the development of numerous strategies to push the limit of grain refinement[3–5]. However, the extension of the grain-size strengthening to the size range below 10–15 nm is restricted either by the challenges in materials generation[6] or by the inverse Hall–Petch effect[7–10]. These challenges are closely related to the size dependence of deformation mechanisms of nanograined metals (NMs). In the grain size range where the Hall–Petch law works, the plastic deformation is determined by the dislocation mechanism. Under mechanical loading, dislocation sources are activated and the glide of lattice dislocations provides the carriers for plastic strain. But when the grain size is further reduced to the so-called extremely-fine nanograined regime with d of a few nanometers, it is suggested that

the grain boundary (GB) processes, e.g., GB sliding and GB migration or coupling[9–11], become dominant in plasticity, which contributes to the observed inverse Hall–Petch effect.

Accessing the strength limit of NMs drives one to explore pathways to tailor the behavior of GBs and dislocations, instead of focusing on the grain size alone. Inspiration is provided by the recently-reported annealing-induced hardening[11–15], which is usually attributed to the GB stabilization by relaxation or segregation. However, for the NMs where the plasticity is clearly mediated by the glide of pre-existing lattice dislocations, the hardening due to the annihilation of such dislocations by annealing is also observed but to a much lesser extent[14]. The potential of strengthening by wiping out lattice dislocations in NMs by annealing is generally limited by the occurrence of significant grain growth, which leads to a net softening of the material.

[1]Herbert Gleiter Institute of Nanoscience, School of Material Science and Engineering, Nanjing University of Science and Technology, Nanjing 210094, China. [2]Institute of Nanotechnology, Karlsruhe Institute of Technology, Karlsruhe 76021, Germany. [3]Institute of Materials Physics, University of Münster, Münster 48149, Germany. [4]Shenyang National Laboratory for Materials Science, Institute of Metal Research, Chinese Academy of Sciences, Shenyang 110016, China. [5]Present address: Key laboratory for Light-Weight Materials, Nanjing Tech University, Nanjing 211816, China. [6]These authors contributed equally: Shangshu Wu, Zongde Kou. ✉e-mail: qingquanlai@hotmail.com; tao.feng@njust.edu.cn

Thus, a question of both fundamental and technological interests arises if design strategies can be devised to create a nanostructure with dislocation-free grain interiors and with GBs that are immobilized and hardened against premature activation of dislocation motion. For that reason, a specific thermo-mechanical processing pathway has been designed to tune the local atomic and chemical structure of the grain interiors and the GBs in a NM. A structural transformation at GBs, which is triggered by low-temperature annealing, exhausts the residual lattice dislocations but avoids significant grain growth. These structural changes could result in the formation of dislocation-exhausted grains and in a transition of plasticity from being dominated by the glide of pre-existing lattice dislocations to being controlled by dislocation nucleation at GBs, which induces an ultra-hardening effect. Considering the rich potential given by different combinations of host alloys and GB-mediated transformations, the present study demonstrates a versatile pathway to explore uncharted regions of mechanical properties of NMs.

## Results

### Nanostructure and mechanical property

We have prepared a nanograined $Fe_{84}Ni_{16}$ (at%) alloy by using the technique of Inert Gas Condensation (IGC). The IGC samples consist of equiaxed nano-sized grains ($15.4 \pm 3$ nm) as shown in Fig. 1a and exclusively of the BCC phase as detected by XRD and TEM (Supplementary Fig. 1). The IGC nanograined Fe-Ni sample exhibits a micro-hardness of $5.1 \pm 0.2$ GPa, which agrees with the reported values for iron and iron alloys of similar grain size[16–18]. Surprisingly, the hardness is dramatically increased upon low-temperature annealing, and the comparison in Fig. 1c shows a peak hardness of $9.4 \pm 0.3$ GPa by annealing at 300 °C for 1 h. No obvious grain growth is observed after such annealing as shown in Fig. 1b. Annealing at higher temperatures results in a softening trend, which goes along with extensive grain growth (Supplementary Fig. 2). Longer annealing duration of the nanograined Fe-Ni alloy at 300 °C shows an even increasing hardness, reaching $10.8 \pm 0.2$ GPa for 10 h (insert in Fig. 1c). In contrast, annealing-induced hardening was not observed for the same alloy with ultrafine-grained (UFG) microstructure obtained by cold rolling. Although the indentation hardness value is correlated to the flow stress at certain plastic strain[19], this technique has been widely used in characterizing the mechanical properties of NMs, and provides a proper ranking of the strength level[3,11,20]. Considering the lack of strain hardening of NMs, the hardness value is regarded as representing the magnitude of yield strength.

The comparison in Fig. 1d clearly highlights the effect of annealing on the IGC Fe-Ni alloy. Firstly, the as-prepared IGC samples present hardness values that fall in line with the Hall–Petch plot of the Fe-based alloys, and the inverse Hall–Petch effect is not observed. This suggests a dislocation-mediated, rather than GB-dominated, plastic deformation, which is also substantiated by the absence of obvious grain growth and grain structure changes in the deformed microstructure underneath the microindents (Supplementary Figs. 3 and 4). Secondly, the hardening efficiency of the IGC Fe-Ni alloy due to annealing has not been achieved in other NMs with $d > 10$ nm (Supplementary Fig. 5). Significant annealing-induced hardening was also reported for the electrodeposited extremely-fine nanograined Ni-Mo alloys with $d < 10$ nm, but the deformation of these alloys is controlled by GB-dominated mechanism that leads to the inverse Hall–Petch effect[11]. Therefore, the present results show a higher hardening effect due to annealing when comparing with other NMs that involve also dislocation-mediated deformation. Furthermore, the IGC-annealed (300 °C for 10 h) Fe-Ni sample with $d = 27 \pm 7$ nm (Supplementary Fig. 6) was even strengthened beyond the level of the extremely-fine NMs ($d$-5 nm), as highlighted in Fig. 1d.

### Structural and chemical analysis

We have performed a systematic characterization campaign by using multiple techniques to reveal the structural and chemical processes during the annealing treatment. The density measurements demonstrate that the initial density of as-IGC nanograined Fe-Ni alloy is -97% of the as-cast sample, and no measurable densification occurs during annealing. An exothermic peak is detected between 100 °C and 200 °C in the DSC curve with an excess energy of 675.8 J/mol (Supplementary Fig. 7), which is comparable to the measurements in refs. 24,25. The relaxation at this stage is probably responsible for the hardness increment from 5.1 GPa to 6.2 GPa by annealing up to 200 °C. Subsequently, atom probe tomography was conducted for the as-prepared IGC and annealed (300 °C for 1 h) nanograined Fe-Ni samples, respectively. Trace impurities (carbon, oxygen, and nitrogen) with a total content of -0.7 at% are found distributing uniformly in the IGC sample and kept unchanged after annealing (Supplementary Fig. 8), which indicates a negligible effect on the annealing-induced ultra-hardening phenomenon. However, the distribution of Ni in the IGC sample is heterogeneous, involving a network-type enrichment and a wavelength comparable to the grain size (Fig. 2a), which is in contrast with the homogeneous distribution of Ni in the cold-rolled UFG Fe-Ni

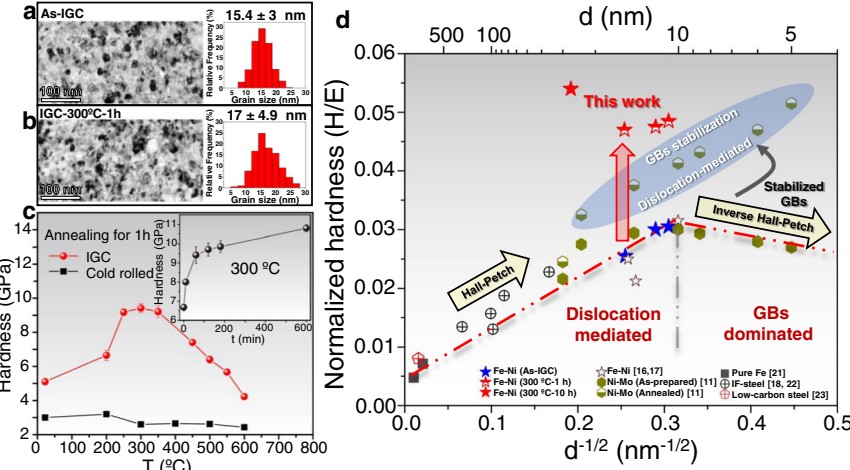

**Fig. 1 | Nanograined Fe-Ni alloy fabricated by IGC and the effect of annealing.** **a**, **b** Bright-field micrograph and grain size distribution of the IGC samples before and after annealing at 300 °C for 1 h. **c** Microhardness as a function of annealing temperature for the IGC nanograined Fe-Ni and cold-rolled Fe-Ni alloys, and the inserted graph shows the microhardness evolution of the IGC sample during annealing at 300 °C. **d** The master plot showing the effect of annealing on the IGC Fe-Ni alloy and the comparison with literature results on Fe-based and Ni-based alloys (Refs. 11,16–18,21–23).

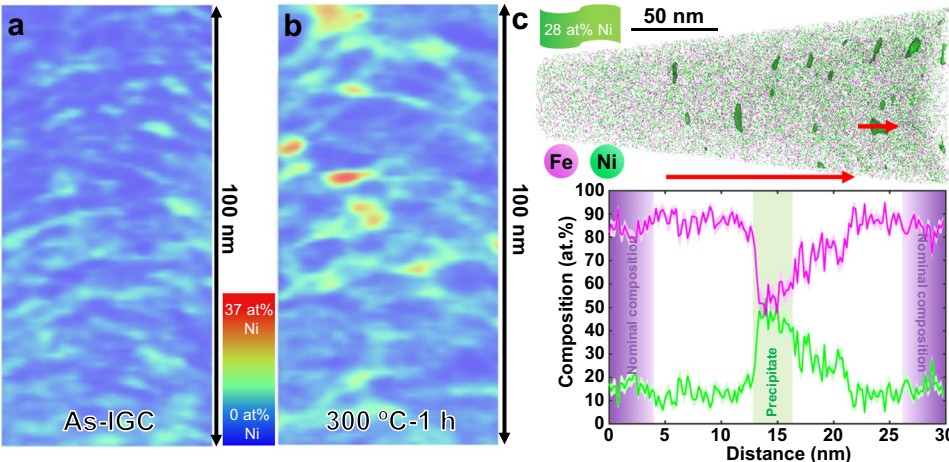

**Fig. 2 | High-resolution chemical analysis of the IGC nanograined Fe-Ni samples. a, b** Three-dimensional reconstruction from atom probe analysis of the IGC and annealed (300 °C for 1 h) nanograined Fe-Ni samples. **c** Quantitative linear analysis of the annealed nanograined Fe-Ni sample.

alloy (Supplementary Fig. 9). The local enrichment of Ni has been found to reach a level of 24 at%. This intergranular Ni enrichment is probably generated during the formation of nanoclusters in the IGC synthesis. Upon annealing, the chemical heterogeneity is enhanced in terms of the size and concentration of the Ni-enriched regions (Fig. 2b). Quantitative analysis of the annealed sample (Fig. 2c) shows that the local Ni concentration increases to 40–50 at%, with a typical size of the regions from a few nm to 15 nm. Additional TEM-EDX mapping also shows the intergranular Ni enrichment, especially at the triple junctions (Supplementary Fig. 10).

The structural changes are investigated by using in-situ synchrotron X-ray diffraction (XRD) during the annealing of the IGC nanograined Fe-Ni alloy (heating to 300 °C and holding for 1 h). Figure 3a and b shows the evolution of the diffractograms, clearly indicating that the single-phase BCC structure is maintained up to 300 °C for short annealing time. During the isothermal annealing, a BCC-FCC transformation occurs, with a content of the FCC phase of 2 vol% at 10 min and 8 vol% at 60 min. Therefore, during annealing, the concurrence of the isothermal BCC-FCC structural change and the diffusion process of Ni re-distribution were observed. The formation of a new phase with a different composition to the matrix suggests that diffusion, most probably along the GB network, plays an important role in the observed structural transformation[26]. Note that 300 °C is a rather low homologous temperature (~0.33$T_m$) for iron. In addition, there is a lack of chemical driving force in Fe–Ni alloy with this nominal composition (Supplementary Fig. 11a). Actually, a much higher FCC-forming temperature (580 °C) is found in the cold-rolled UFG counterpart (Supplementary Fig. 11b). However, the unique structural and chemical features of the IGC nanograined alloy facilitate the onset of the transformation. Firstly, the local Ni enrichment tends to increase the thermodynamic driving force[27] and favor the formation of FCC phase by decreasing the BCC-FCC transformation temperature (Supplementary Fig. 11a). Secondly, the enhanced diffusivity due to the high density of high-angle GBs kinetically facilitates the transformation processes[28]. Assuming the width of GBs to be 0.5 nm, the GB diffusion distance of Ni in BCC Fe at 300 °C for 1 h is estimated as 22 nm[29–31], which is a value comparable to the grain size. Thirdly, the local compressive strain field at the GB regions, as revealed by the geometrical phase analysis in Supplementary Fig. 12, could help accommodate the volume shrinkage during the BCC-FCC transformation and provide additional internal mechanical driving force[32,33].

The diffraction profile analysis provides the critical structural parameters, including the average grain size and the magnitude of micro-strain distribution, according to the modified Williamson–Hall

method[34]. The details of the profile analysis can be found in Supplementary Fig. 13, and the results of the analysis are listed in Supplementary Table 1. As shown in Fig. 3d, a high initial density of lattice dislocations is observed in the as-prepared IGC state (1.2 × 10^15 m^-2), which decreases during annealing. After annealing at 300 °C for 1 h, the dislocation density is substantially reduced to 4.1 × 10^13 m^-2. A salient increase in hardness is observed accompanied with the decrease of dislocation density.

The structural changes occurring during annealing are examined in detail using HR-TEM. In the IGC sample, we clearly observe dislocations adjacent to the GBs as shown in Fig. 4a. Occasionally, dislocations are also observed that are located in the center of the grains (Supplementary Fig. 14a). In addition, the GPA results show that the lattice dislocations are also inducing a long-range stress field (Supplementary Fig. 12)[32,33]. Besides the edge dislocations revealed directly by HR-TEM, the rigid body rotation also indicates the presence of screw dislocations (Supplementary Fig. 12). In the annealed sample, the dislocation annihilation creates the dislocation-exhausted BCC nanograins, a typical example being shown in Fig. 4b. The FCC phase with a size of ~5 nm nucleated at the GBs. The observed significant decrease in dislocation density and the structural transformation are consistent with the synchrotron diffraction results. When deforming the annealed sample, significant dislocation storage occurs (as shown in Supplementary Fig. 14b), which again suggests the occurrence of dislocation-mediated plastic deformation processes in the present nanograined system.

For dislocation-mediated plasticity in NMs, it is usually assumed that the grain interior is free of dislocations, and dislocations are nucleated at the GB, slipping through the entire grain and being absorbed by the GB on the opposite side of the grain[35–38]. However, the pre-existence and storage of dislocations in nanograins are issues under debate, and the dislocation-free assumption in the above physical picture has been challenged by a series of studies using X-ray line profile analysis and/or HR-TEM[15,39,40]. The significant peak broadening in the diffraction patterns of the nanograined alloy is not only caused by the very small grain size but also by the inhomogeneous internal elastic strains due to the existence of lattice dislocations. Note that only dislocations remaining in the lattice cause the long-range stress fields, which is clearly shown in Supplementary Fig. 12, and induce peak broadening, while dislocations absorbed in the GBs are not associated with such long-range stress fields[39]. In the as-prepared IGC samples, the dislocation density of the order of 10^15 m^-2 is comparable to the reported magnitude in other NMs prepared by severe plastic deformation[40,41]. In the IGC samples, dislocations are located near the

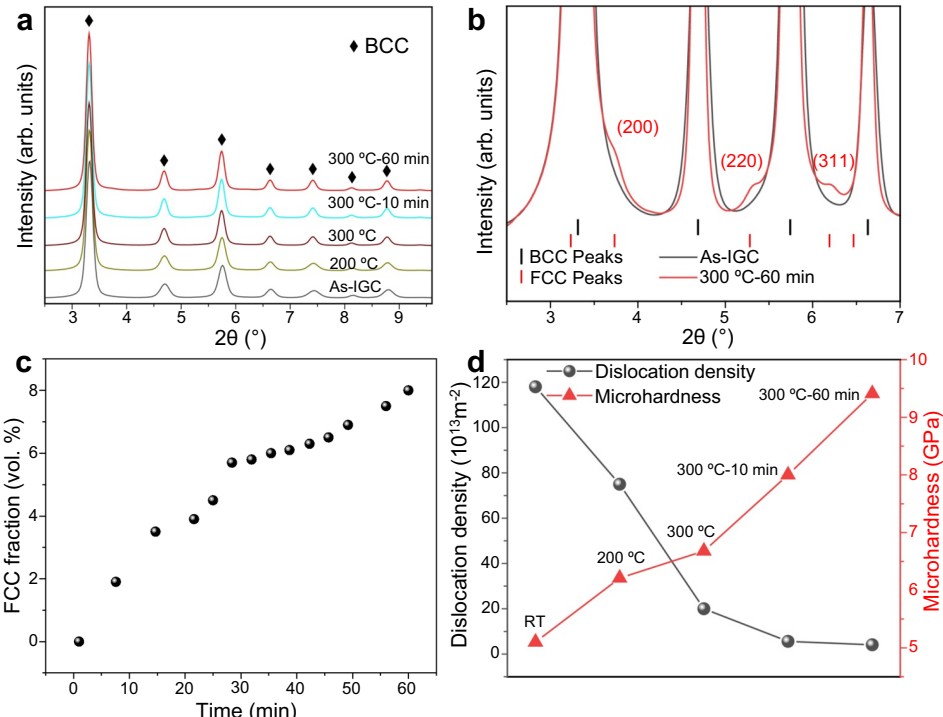

**Fig. 3 | In-situ synchrotron XRD results of the IGC nanograined Fe-Ni samples. a, b** Synchrotron XRD profiles of the IGC Fe-Ni sample. **c** Evolution of the fraction of FCC phase during isothermal heat treatment at 300 °C. **d** Variations of dislocation density and microhardness of the nanograined Fe-Ni sample.

GBs as shown in Fig. 4a. Such lattice dislocations attached to GBs can be activated under the applied stress and consequently accommodate plastic deformation. After annealing at 300 °C, the dislocation density is reduced by two orders of magnitude to ~$10^{13}$ m$^{-2}$, close to the level of well-annealed metals[42]. In severely-deformed metals, such magnitude of dislocation annihilation can only be achieved by annealing at a high homologous temperature[42]. The above measurements suggest that the lattice dislocations in the present nanograins are exhausted by annealing, and alternative dislocation sources have to be activated at the GBs, which requires a higher stress and induces the dramatic hardening observed in the present study.

To elucidate the significance of the GB-mediated transformation, we establish the linkage between phase transformation, dislocation density, and hardness as shown in Supplementary Table 1. During the heating period from RT to 300 °C without phase transformation, the dislocation density decreases from $1.2 \times 10^{15}$ m$^{-2}$ to $2.0 \times 10^{14}$ m$^{-2}$ (83% reduction) associated with a hardness increase from 5.1 GPa to 6.7 GPa (31% increment). The larger hardness increase of 53% (from 6.7 GPa to 9.4 GPa) is associated with the phase transformation during the holding stage, accompanied by an additional 13% reduction of the dislocation density from $2.0 \times 10^{14}$ m$^{-2}$ to $4.1 \times 10^{13}$ m$^{-2}$. Although the phase transformation does not result in a sharp decrease of the dislocation density in an absolute sense, it does change the nature of the dislocation structure in the grains, i.e., by wiping out the residual lattice dislocations to a very low level. The significance of the residual lattice dislocations can be highlighted by the estimation that the sweeping of a single dislocation in a nanograin with $d$~15 nm can produce a plastic shear of the order of $b/d$ (~1.6%)[43]. When the intragranular dislocation sources were exhausted after the occurrence of phase transformation, the dislocation activation becomes controlled by GB sources.

## Discussion

The IGC nanograined Fe-Ni alloy involves a grain size of ~15 nm, which is within the grain size regime where the Hall–Petch law works, without involving the inverse Hall–Petch effect. The absence of stress-driven grain growth was evidenced, indicating that the GB migration is not a

significant deformation mechanism. These experimental observations demonstrate a dislocation-mediated deformation mechanism. A high density of pre-existing lattice dislocations was identified by the XRD profile analysis. And the ultra-hardening is suggested to be attributed to the dramatic reduction of lattice dislocation density via phase transformation at GBs during annealing.

The mechanism of dislocation exhaustion by such a structural transformation can be explained as follows: In the nanograined structure produced by the IGC process, the lattice dislocations attached to the GBs are enhancing the local internal energy through the contribution of strain energy. In addition, the lattice dislocations at the GB region are presumably increasing the diffusivity. Thus, the locations of GBs with attaching dislocations provide the favored nucleation sites for the structural transformation. In return, the transformation events consume such lattice dislocations attaching to the GBs. In addition, the transformation involves a flux of diffusing species (vacancies, atoms), which could accelerate the annihilation of the residual lattice dislocations by, e.g., dislocation climb. These effects are suggested to facilitate the formation of a dislocation-exhausted nanograined structure.

Since the small FCC islands are formed at the GBs of the BCC matrix, they are not reducing the mean free path of dislocation glide as in the case of precipitation hardening. In addition, the FCC second phase is not expected to be responsible for such ultra-hardening effect through the composite strengthening mechanism. If we estimate the hardness of the second phase according to the rule of mixtures, a value of 55 GPa is obtained, which is too high for a metallic phase constituent. We have also prepared a FCC $Fe_{50}Ni_{50}$ nanograined sample (grain size: $21 \pm 5.5$ nm) by IGC (see Supplementary Fig. 15) and obtained a hardness of 4.2 GPa, which is lower than the hardness of the BCC $Fe_{84}Ni_{16}$ sample and is thus not supposed to act as a reinforcement. In order to experimentally substantiate the proposed GB-mediated transformation strengthening pathway, we prepared a nanograined Pd-Au alloy with $d = 19.2 \pm 6$ nm using IGC, which is a solid solution without phase transformation during annealing. In this case, we observe a moderate hardening of 0.75 GPa ($\Delta H_V/H_{V0} = 17.6\%$) by

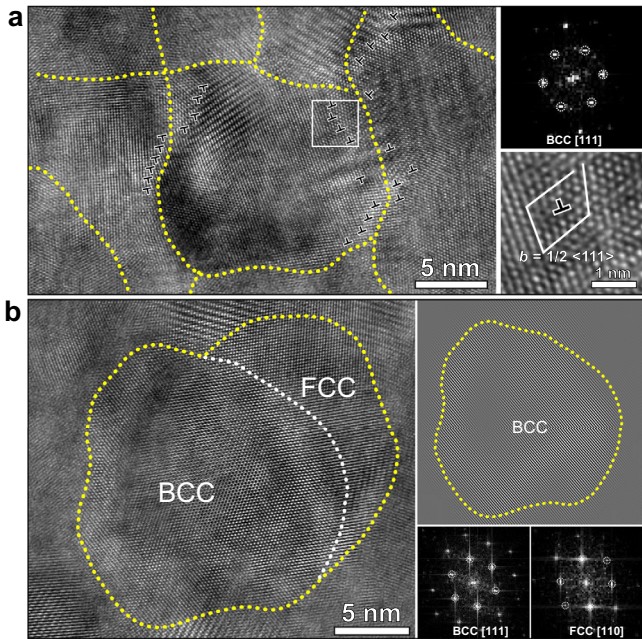

**Fig. 4 | HRTEM observations of the IGC nanograined Fe-Ni samples before and after annealing at 300 °C for 1 h. a** This shows the dislocations (marked by T shape symbols) at the grain boundary region (marked by the dotted yellow line), FFT and a typical burgers circuit of the Fe-Ni sample before annealing. **b** HRTEM image of the 300 °C annealed Fe-Ni sample, showing the coexistence of BCC (electron beam is along the [111]) and FCC (electron beam is along the [110]) phases, and a dislocation-free nanograin.

annealing when compared with the as-prepared IGC state, which is induced by mechanisms other than phase transformation (Supplementary Fig. 16).

It should be noted that the ultra-hardening effect observed here is also attributed to the moderate stability of grain size, since significant grain growth could counteract and overwhelm the strengthening contributions. The grain size of 15 nm was maintained during the annealing at 300 °C for 1 h, while annealing at a higher temperature (e.g., 500 °C) results in an increased grain size (~90 nm) and in a reduced hardness. The thermal stability of the present nanograined Fe-Ni alloy is higher than the nanograined pure iron prepared also by IGC[44]. This is probably attributed to the Ni segregation and the formation of FCC phase at the GBs during the low-temperature annealing, which pin the GB migration and retard grain growth. Note that the substitutional Ni strengthens the lattice of BCC iron[45], and the Ni depletion in the BCC phase due to the formation of Ni-rich FCC phase tends to reduce the shear resistance, which was not observed in the experiments. This observation can be rationalized as follows: when the plastic deformation is controlled by dislocation nucleation at GBs, the strengthening due to a high defect nucleating stress will overwhelm the reduction of solid solution hardening.

The present study demonstrates that the strength of the NMs is determined not only by the grain size and GB stability, but also by the quantity of the residual lattice dislocations. When decreasing the grain size to the nanometric scale, the intragranular dislocation sources (e.g., Frank-Read source) no longer prevail, but the residual lattice dislocations could still act as carriers of plasticity without the necessity to excite GB dislocation sources. As mentioned before, the strengthening potential by dislocation annihilation in NMs is still experimentally unclear[15], and it is easily overwhelmed by the significant grain growth. Here, in our present approach of engineering the physicochemical characteristics of GBs in the nanograined alloys, we can trigger the phase transformation at GBs at a low temperature to highly

exhaust the residual lattice dislocations. The plastic deformation is then suggested to be dominated by the dislocation emission from GBs, which allows for approaching the strength limit of a given nanograined polycrystal. As shown in Fig. 1d, a comparable strength can be achieved by further reducing the grain size to the regime of extremely-fine nanometric grains (d~5 nm) when the GBs are stabilized, the plastic deformation of which is also controlled by dislocation emission from GBs. In addition, when the plasticity is determined by dislocation emission from GBs, the ultra-high strength can be achieved within a range of grain size, associating with a less stringent requirement of grain refinement. The theoretical shear strength, for either the shearing of two neighboring planes[46] or the homogeneous nucleation of a dislocation[47], could be estimated as 6.6 GPa (with the order of G/10) for BCC iron. This corresponds to a Vicker's hardness of ~60 GPa when assuming a random crystal orientation and using the empirical relationship between strength and hardness, which is several times higher than the observed maximum hardness in this study (10.8 GPa). But the measured ultrahigh hardness is considered as approaching to the magnitude of the theoretical limit, when accounting for the influences of the reduced elastic modulus by Ni addition and the possible stress concentrators. The as-prepared IGC nanograined Fe-Ni alloy is initially decorated by the intergranular Ni enrichment, but the hardness is comparable to the iron alloys with similar grain sizes (Fig. 1d), which is not suggesting the primary and direct contribution of the GB segregation to the ultra-hardening phenomenon. However, the further Ni segregation at GBs and the structural transformation during annealing could induce other consequences. The GB structure could be changed by the phase transformation at GBs, which might influence the dislocation nucleation stress[14,15,48]. Since the non-equilibrium locations at the GBs are favored sites for both the transformation and the dislocation nucleation[49,50], the formation of FCC phase at such locations is increasing the apparent dislocation nucleation stress. This effect should be more significant when the preexisting lattice dislocations are highly exhausted and when the GB characteristics dominate the onset of plasticity, which is relevant to the continuous hardening during the prolong annealing at 300 °C (Fig. 1c).

We have demonstrated a capability to control phase transformation for development of engineering materials. It is still challenging, but promising to extend this approach to the development of nanograined materials. Controlling phase transformations in the nanograined materials through engineering the thermodynamic and kinetic properties of GBs and interfaces, as is exemplified in this study, opens a new door to the development of nanostructures with desired properties.

## Methods

### Sample preparation

The Fe-Ni nanograined alloy was synthesized by the Inert Gas Condensation (IGC) system with thermal evaporation method. The IGC system consists of a powder preparation unit, a low-pressure compaction unit, a high-pressure compaction unit, and the base vacuum of the system is $10^{-6}$ Pa. These units are interconnected by an ultra-high vacuum transfer line. The raw material involves a composition of $Fe_{75}Ni_{25}$. Before preparing the sample, the vacuum of the powder preparation unit was $<10^{-6}$ Pa, and then a low-pressure (500 Pa) inert gas He was introduced. The liquid nitrogen was introduced into the cold finger, and the sample was evaporated into a gaseous state by resistance heating. The metal atoms lost energy after colliding with He gas, and deposited onto the cold finger. On the cold finger, the powder was scraped off and collected into a mold. After being transferred to the low-pressure compaction unit, the powders are compressed into a pellet with 10 mm diameter at an applied uniaxial pressure of approx. 500 MPa. After this pre-compaction step, the pellet was transferred to the high-pressure compaction unit where a pressure of 5 GPa was applied for 3 min at ambient temperature. Finally, we get the $Fe_{84}Ni_{16}$

**Table 1 | The values of $\bar{C}$ for different crystallographic plane**

| hkl | Screw | Edge | Half screw + half edge |
|-----|-------|------|------------------------|
| 110 | 0.08619 | 0.17472 | 0.1305 |
| 220 | 0.26 | 0.256 | 0.258 |
| 211 | 0.08619 | 0.17472 | 0.1305 |
| 220 | 0.08619 | 0.17472 | 0.1305 |
| 310 | 0.1974 | 0.2267 | 0.21205 |

nanograined alloy with thickness of 300 μm. By adjusting the pressure of He gas (200 Pa, 50 Pa) in the preparation process, we obtained samples with smaller grain sizes. As a comparison, the sample of $Fe_{84}Ni_{16}$ alloy was prepared by arc melting, and then was cold-rolled by 90% for the ultrafine-grained microstructure.

The IGC Fe-Ni samples were annealed at modest temperatures (200 °C, 250 °C, 300 °C, 350 °C, 450 °C, 500 °C, 550 °C, 600 °C) for 1 h (with a heating rate 20 °C/min) under the protection of an argon atmosphere.

### Microhardness measurement
Microhardness measurements were performed using a Vickers microhardness tester with a load of 200 g and a dwell time of 10 s. More than ten indents were made for each condition.

### Structural characterization
TEM and HRTEM were conducted on a Themis Z microscope operated at 300 kV. TEM and HRTEM specimens before and after the micro-hardness indentation were prepared by using FEI Helios Nanolab 600i system operated at a voltage of 30 kV to process the sample to -1000 nm, then reduce the voltage to 16 kV and continue to thin to -500 nm, reduce the voltage to 8 kV and continue to thin to -200 nm, use 5 kV to reduce to -100 nm, and finally reduce the voltage to 2 kV and gradually reduce the thickness of the sample to -50 nm. After thinning to the target thickness, clean the sample with a low voltage of 1 kV to remove the amorphous layer introduced by the ion beam damage. TEM specimen of the cold-rolled Fe-Ni was mechanically ground to 50 μm in thickness, followed by final thinning using dimpler (GrinderII, Model 657) and ion milling (PIPSII, Gatan 695). The average grain sizes were measured from the bright-field TEM images by using the Heyn intercept method[51]. More than 400 grains were measured on each sample. In the measurement of grain size distribution, the grains for the FCC and BCC phases were not distinguished.

The IGC Fe-Ni alloy was characterized by using the in-situ synchrotron high-energy X-ray diffraction at beamline 11-ID-C at the Advanced Photon Source, Argonne National Laboratory. High-energy monochromatic X-rays with a beam size of 500 μm × 500 μm and a wavelength of 0.1173 Å were used in transmission geometry for data collection. The sample was heated from room temperature to 300 °C at a rate of 20 °C/min and then kept at 300 °C for 1 h.

### Chemical analysis
The chemical compositions of the IGC Fe-Ni samples were determined by the energy dispersive spectroscopy on a Quant 250FEG operated at 15 kV. Specimens for APT were prepared on a FEI strata FIB. The APT experiments were conducted on a Cameca LEAP 4000 XHR. The APT data reconstruction and post-processing was done with the commercial software IVAS 3.8.4[52].

### Differential scanning calorimeter measurement
Differential scanning calorimeter (DSC) measurements were performed with a Mettler Toledo DSC1. Approximately 20 mg of samples were sealed in Al pans and scanned at a heating rate of 20 °C/min from room temperature to 300 °C. An empty Al pans was tested under the same conditions in order to determine the baseline.

### Determination of phase transformation of cold-rolled Fe-Ni
Thermal standard expansion measurements were conducted in the compression mode using NETZSCH 402F3 thermal mechanical analyzer. A constant pre-set load of 10 mN was applied on the samples during the entire measurement. The samples were protected by the nitrogen flow with 20 ml/min, and the temperature ranges from 27 °C to 800 °C.

### Measurement of dislocation density
The modified Williamson–Hall (MWH) method was used for analyzing the X-ray diffraction line broadening and measuring the dislocation density. The full widths at half maximum (FWHM) of the diffraction profiles are determined as the widths of Gaussian and Cauchy functions fitted to the experimental diffraction data[53]. Instrumental broadening is calibrated based on the values of FWHM of the standard sample ($CeO_2$).

For MWH method, the FWHM of diffraction peak profiles are plotted versus $K\bar{C}^{1/2}$ as given by refs. [34],[54]:

$$\Delta K = 0.9/D + (\pi A^2 b^2/2)^{1/2} \rho^{1/2}(K\bar{C}^{1/2}) + O(K\bar{C}^{1/2})^2 \quad (1)$$

where $D$ and $K$ represent the apparent size parameters and the diffraction vector ($K = 2\sin\theta/\lambda$, $\theta$ is the Bragg angle, $\Delta K = 2\Delta\theta\cos\theta/\lambda$ and $\lambda$ is the X-ray wavelength). $\bar{C}$ (Table 1) is the average contrast factor of dislocations, and $O$ is a higher order term of $K\bar{C}^{1/2}$. $A$ is constant depending on the outer cut-off radius of dislocations, $b$ is the Burgers vector of dislocations.

In a polycrystalline cubic metal, the average values of the contrast factors are calculated by[55]:

$$\bar{C} = \bar{C}_{h00}(1 - qH^2) \quad (2)$$

where $\bar{C}_{h00}$ is a constant depending on the anisotropic elastic constants $C_{11}$, $C_{12}$ and $C_{44}$, $q$ is a dislocation parameter recognizing relative fractions of screw and edge type dislocations[56]. For a given ($hkl$) reflection, $H^2$ is determined by[55]:

$$H^2 = (h^2k^2 + k^2l^2 + l^2h^2)/(h^2 + k^2 + l^2) \quad (3)$$

The value of $d$ and $\rho^{1/2}$ can be determined by the best linear fit between $K$ and $\Delta K$.

### Assessment of the thermodynamic properties of the Fe-Ni alloy
The thermodynamic properties of the Fe-Ni binary system have been established in ref. [27]. The difference between the chemical free energy of the FCC and BCC phases in Fe-Ni binary alloys can be expressed as:

$$\Delta F^{BCC \to FCC} = (1 - x)(1202 - 2.63 \times 10^{-3}T^2 + 1.54 \times 10^{-6}T^3) + x(-3700 + 7.09$$
$$\times 10^{-4}T^2 + 3.91 \times 10^{-7}T^3) + x(1 - x)[3600 + 0.58(1 - \ln T)](\text{cal/mol}) \quad (4)$$

where $x$ is the atomic fraction of Ni and $T$ is the absolute temperature. Thus, $T_0$ temperature can be determined when $\Delta F^{BCC \to FCC}$ equals to zero. And the chemical driving force for the BCC-FCC structural change at a given temperature beyond $T_0$ can also be provided.

## Data availability
The data that support the findings of this study are available from the corresponding author upon request. Source data are provided with this paper.

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

## Acknowledgements

The authors thank Prof. K. Lu for his insightful and constructive comments on this paper. We thank Dr. J. J. Wang and Dr. M. Y. Yan for the help with sample preparation, Dr. S. N. Liu and H. Q. Ying for the synchrotron XRD test and analysis, Dr. S. Fu for the help with Rietveld refinement. Q.Q.L. acknowledges National Natural Science Foundation of China under grant number 52001166. T.F. acknowledges support from National Natural Science Foundation of China under grant numbers 51520105001 and 51571119, Fundamental Research Funds for the Central Universities under grant number 30919011404, Qing Lan project of Jiangsu province and Distinguished professor project of Jiangsu province. Z.D.K. acknowledges Natural Science Foundation of Jiangsu Province under grant number BK20210352. S.L. acknowledges National Natural Science Foundation of China under grant number 51871120, the National Key R&D Program of China (No. 2021YFB3802800), Natural Science Foundation of Jiangsu Province under grant number BK20200019. This research used the resources of the Advanced Photon Source, a US Department of Energy (DOE) Office of Science User Facility operated for the DOE Office of Science by Argonne National Laboratory under Contract No. DE-AC02-06CH11357, and was supported by the US DOE Office of Science, Office of Basic Energy Sciences.

## Author contributions

T.F. and Q.Q.L. designed the research. S.S.W. and Z.D.K. prepared and characterized the samples. Z.D.K., S.L., S.S.K., and S.T. conducted the TEM, synchrotron XRD, 3D-APT, and TEM-EDX analyses, respectively. S.S.W., Z.D.K., Q.Q.L., H.H., G.W., T.F., and H.G. analyzed the data and discussed the results. S.S.W., Q.Q.L., H.H., G.W., T.F., and H.G. wrote the manuscript. All authors reviewed and contributed to the final manuscript.

## Competing interests

The authors declare no competing interests.
