## [Peer Review File · Nature Communications]

Title: Dislocation exhaustion and ultra-hardening of nanograined metals by grain-boundary mediated transformationsREVIEWER COMMENTS

Reviewer #1 (Remarks to the Author):

The research reported in this manuscript is extremely interesting and will be for sure of interest for a large scientific community. The authors have designed a unique nanograined metallic alloy that can be strengthened by annealing. They carried out intensive and very detailed structural analyses using various techniques to collect data that are used to reveal the underlying mechanisms. That way, they demonstrate that hardening was produced by creating a "dislocation free" two phases nanostructure without obvious grain growth. Thus, the authors conclude that the observed hardening is the result of the necessary activation of dislocation sources at GBs.

The paper is well written and well organized. The experimental data are of high quality and previous work in the field is appropriately quoted. Many experimental details are provided making data interpretations very convincing.

The reviewer would like to indicate several points that might be improved or clarified before publication :

- The authors claim that impurities (C, O, N) are homogeneously distributed (op of page 5), it would be nice to show corresponding 3D chemical maps in extended data because these impurities could affect the grain boundary mobility and also give rise to a rather limited grain growth.
- Page 6 line 11, the authors discuss about the diffusion of Ni and the formation of the fcc phase. It occurred at relatively low temperature and they argue that atomic mobility may be enhanced by crystalline defects. The GB diffusion coefficient of Ni in Fe is probably known, thus it would be nice to check the expected diffusion distance of Ni along GBs during 1h at 300°C. Such estimate would make the discussion more convincing.
- Page 6, line10, it is not clear why a local enrichment is going to change the thermodynamic driving force for phase separation, please clarify.
- Page 6, lines 14-15 : note that the density is about 97% (page 4, line 27), so volume expansion is probably not a problem at all
- The authors never discuss about the solid solution hardening, this is quite surprising because they have to admit that the formation of the fcc Ni rich phase should lead to a Ni depletion of the Fe bcc phase and consequently to a reduction of the solid solution hardening.

Reviewer #2 (Remarks to the Author):

The manuscript by Wu et al. reported a ultra-hardening phenomenon in nanograined metals. By introducing grain segregation and dislocation-free grain structures, nanograined Fe-Ni alloy with d of tens of nanometers were strengthened to a ultrahigh level. Multiscale characterizations have been applied to reveal the microstructural origin of ultrahigh hardening, which was attributed to the activation of dislocation sources at GBs in the dislocation-free nanograins. Though this work is interesting, major clarifications need be further addressed, especially regarding the origin of observed-

phenomenon:

1. In introduction, pre-existing dislocations are not always the dominating factor of the plasticity of nanograined metals. It is structure dependent. And the softening is mainly induced by the strain localization rather than grain growth.
2. How was the grain size measured? It is necessary to provide the grain size distributions for Fig. 1a-b. Also, what is the grain size of UFG samples? More detailed TEM characterization needs to be provided to show the structure of different samples.
3. Page 4. I think it is kind of difficult to exclude the GB mechanism based on current results. In Extended Figure 3, some large grains seem formed beneath the indentation surface? It is better to provide some zoomed-in dark field image. And what are the grain structures under the side surfaces of indentation? The deformation can be different with large shear stress.
4. Fig. 2 cannot represent the typical structural feature of the annealed structures. The GBs with Ni segregations should play more important roles on the hardening, rather than the cluster with high Ni concentration. Similar behaviors have been widely reported in literatures.
5. Are there any contribution of impurities to the hardening? How to exclude its contribution? The impurity content of 0.7 at% is not so low, and thus needs to be discussed.
6. The so-called Phase transformation may not be appreciated to describe the structural transformation during annealing. It is a process similar to the precipitation or GB aggregation of Ni-enriched clusters. And also, the contribution of so-called phase transformation to hardening should be low. The hardening should be mainly originated from the reduced GB mobility due to the GB pinning, as well as the change of defect nucleation stress at aggregated GB, which has been well-studied in literatures.
7. What is the diffusional transformation really mean? A clear definition/description with more evidences need to be provided. Complex GB processes and element redistribution should be involved in this process. The lack of grain growth also needs to be discussed along with this diffusional transformation.
8. The hardening reported in this paper was simply attributed to dislocation-free-induced GB nucleation. Based on my knowledge, several other factors should contribute to the high hardness reported in this paper, including low dislocation density, GB segregation and the resultant reduced GB mobility, as well as the segregation-induced change of dislocation nucleation stress from GB. The change of GB dynamics should play more important roles.
9. "Dislocation-free" cannot represent the main structural feature of the annealed samples, which needs to be changed in both abstract and the main text.
10. Discussion needs to be further improved in the context of the existing literature in an appropriate manner.

Reviewer #3 (Remarks to the Author):

The manuscript shows that annealing at moderate temperatures dramatically increases the hardness of an Inert Gas Condensation produced Fe-Ni alloy. The authors argue that this is due to diffusional

transformations at the grain boundaries that wipe out the existing dislocations. The resulting dislocation-free structure would hinder the nucleation of follow-up dislocations and account for the strength increase. This theory is interesting, but insufficiently supported by direct evidence.

1. Hardness measurements are not suited to measure the incident yield strength (which indeed would be increased by hindering dislocation nucleation), because they introduce a large amount of plastic straining, so the reported hardness corresponds to a flow strength after multiple dislocations have presumably already been emitted and the restriction on dislocation nucleation long overcome.
2. The second issue is that the authors are reaching their conclusions by ruling out alternative explanations rather than direct evidence, see below. If the dislocation density would be the main factor, why not rationalize the increase in strength by a dislocation model – or at least provide the theoretical strength of Fe-Ni for comparisons to the experimental values?
3. The conclusions are based on the assumption of a stable microstructure. This critical point is insufficiently supported by data. The authors must provide the full grain size distribution both BEFORE and after heat treatment at 300°C. To me, it appears that the proportion of smaller grains must increase, since <5 nm large fcc grains are nucleated (see Fig. 4b).
4. My own feeling is that the increase of hardness is rather connected to the nucleation of this fine-dispersed (see Fig. 2c as well as the fine lamellar structure in ExFig 7a) phase and to the juxtaposition of bcc/fcc phases, than from the dislocation-free condition.

Further remarks:

5. The authors are not discussing the influence of the residual 3% porosity and 0.7% impurities, which is also a major difference to the cold-rolled sample and might also play a role in the phase transformation.
6. It is not clear to me how the authors can experimentally quantify and distinguish the grain boundary dislocations from the grain-interior dislocations.
7. The references are split in two by the methods section, which makes them very inconvenient to read.

Report of reviewer 1-- NCOMMS-22-02265A

Reviewer #1 (Remarks to the Author):

The research reported in this manuscript is extremely interesting and will be for sure of interest for a large scientific community. The authors have designed a unique nanograined metallic alloy that can be strengthened by annealing. They carried out intensive and very detailed structural analyses using various techniques to collect data that are used to reveal the underlying mechanisms. That way, they demonstrate that hardening was produced by creating a "dislocation free" two phases nanostructure without obvious grain growth. Thus, the authors conclude that the observed hardening is the result of the necessary activation of dislocation sources at GBs. The paper is well written and well organized. The experimental data are of high quality and previous work in the field is appropriately quoted. Many experimental details are provided making data interpretations very convincing.

The reviewer would like to indicate several points that might be improved or clarified before publication:

General Response: The authors are grateful to the reviewer for his/her positive comments and insightful summaries of work. Based on these comments, we have improved the manuscript for better clarity.

- The authors claim that impurities (C, O, N) are homogeneously distributed (op of page 5), it would be nice to show corresponding 3D chemical maps in extended data because these impurities could affect the grain boundary mobility and also give rise to a rather limited grain growth.

Response: In the revised version, the 3D chemical maps of the impurities for the nanograined Fe-Ni sample before and after annealing are provided in the Supplementary Fig. 8. Here, as shown below in Figure R1, the impurities (C, O, N) are homogeneously distributed within the sample, and segregation and re-distribution are not observed after annealing at 300 °C for 1 h. Comparing with the obvious re-distribution of Ni atoms during annealing (as shown in Fig. 2), the unaltered distribution of the impurities indicates that it is not significant in the grain boundary mobility and grain growth.

Figure R1. The 3D reconstruction of impurities (C, O, N) in the Fe-Ni nanogained alloy (a) before and (b) after annealing at 300 °C for 1 h.

Changes to the manuscript: In the revised manuscript, we have provided the 3D chemical maps of the impurities (C, O, N). The modified parts have been highlighted in red.

Page 5, Line 130-131, main text.

Page 9, Newly added Supplementary Fig.8, Supplementary Information.

- Page 6 line 11, the authors discuss about the diffusion of Ni and the formation of the fcc phase. It occurred at relatively low temperature and they argue that atomic mobility may be enhanced by crystalline defects. The GB diffusion coefficient of Ni in Fe is probably known, thus it would be nice to check the expected diffusion distance of Ni along GBs during 1h at 300°C. Such estimate would make the discussion more convincing.

Response: The GB diffusion of Ni in Fe is a classical topic in metallurgy. A nice set of data on the lattice and GB diffusion of Ni in Fe can be found in Ref^[1]. The lattice diffusion coefficient of Ni in α -Fe, D_b , is given by:

$$D_b = 1.4 \times \exp(-58700 \times 4.19 / RT) \text{ cm}^2 / \text{s} \quad (1)$$

The grain boundary (GB) diffusion coefficients of Ni in α -Fe, D_{GB} , is provided by:

$$D_{GB} \delta = 23.3 \times 10^{-7} \times \exp(-43300 \times 4.19 / RT) \text{ cm}^3 / \text{s} \quad (2)$$

where R is the gas constant 8.314 J/(mol·K), T is the absolute temperature and δ is the GB width that is estimated as 0.5 nm^[2]. Therefore, the GB diffusion distance ($\sqrt{D_{GB}t}$)^[3] at 300 °C for 1h is ~22 nm, which compares well with the grain size of the

IGC Fe-Ni alloy. On the contrary, the lattice diffusion distance ($\sqrt{D_b t}$) at 300 °C for 1h is $\sim 4.4 \times 10^{-3}$ nm, which is a value much lower than an atomic jump. Therefore,

this comparison substantiates quantitatively the argument that the low-temperature diffusional transformation is facilitated by the enhanced atomic mobility due to the high density of GBs.

References

- [1] D.W. James, G.M. Leak, Grain boundary diffusion of iron, cobalt and nickel in alpha-iron and of iron in gamma-iron, Philosophical Magazine 12, (1965) 491-503.
- [2] S. Divinski, F. Hisker, Y.S. Kang, J.S. Lee, C. Herzig, Tracer Diffusion of ^{63}Ni in Nano- γ -FeNi Produced by Powder Metallurgical Method: Systematic Investigations in the C, B, and A Diffusion Regimes, Interface Science 11, (2003) 67-80.
- [3] S. Divinski, H. Rösner, G. Wilde, Chapter 1-Functional Nanostructured Materials–Microstructure, Thermodynamic Stability and Atomic Mobility, Frontiers of Nanoscience 1, (2009) 1-50.

Changes to the manuscript: In the revised manuscript, we have provided the estimated diffusion distance of Ni along GBs during 1 h at 300°C. The modified part has been highlighted in red.

Page 7, Line 168-170, main text.

- Page 6, line10, it is not clear why a local enrichment is going to change the thermodynamic driving force for phase separation, please clarify.

Response: The thermodynamic properties of the Fe-Ni binary system is well established. And the driving force for the BCC \rightarrow FCC structural change can be calculated based on the chemical free energy of the FCC and BCC phases as a function of the Ni content and temperature^[1]. The chemical free energy change accompanying the structural change in the Fe-Ni system is shown below in Figure R2 (see also Supplementary Fig. 11a). The T_0 temperature is denoted as the equilibrium temperature. At temperatures higher than T_0 , the formation of the FCC phase is energetically favored. The T_0 for $\text{Fe}_{90}\text{Ni}_{10}$, $\text{Fe}_{84}\text{Ni}_{16}$ and $\text{Fe}_{80}\text{Ni}_{20}$ are 570 °C, 425 °C and 370 °C, respectively, which decreases with an increasing Ni content. In addition, the chemical free energy change at a given temperature beyond T_0 is also increased by a higher Ni content. Therefore, the local Ni enrichment tends to favor the formation of the FCC phase by decreasing the T_0 temperature and also tends to increase the thermodynamic driving force.

Figure R2. Chemical free energy change accompanying the BCC→FCC structural change in the Fe-Ni system (x is the atom fraction of solute Ni in the Fe-Ni alloy).

References

[1] L. Kaufman, M. Cohen, Thermodynamics and kinetics of martensitic transformations, *Prog. Metal. Phys.* 7 (1958) 165-246.

Changes to the manuscript: In the revised manuscript, we have improved the discussion on the enhanced thermodynamic driving force for phase transformation derived from the local Ni enrichment. The modified parts have been highlighted in red. Page 6, Line 164-165, main text.

Page 7, Line 166, main text.

Page 17, Line 441-442 and 446-449, methods.

- Page 6, lines 14-15: note that the density is about 97% (page 4, line 27), so volume expansion is probably not a problem at all

Response: On Page 6, we focused on the discussion of the possible mechanisms favoring the low-temperature phase transformation. For pure α -Fe, the volume shrinkage during the BCC→FCC phase transformation is 8.14%. Thus, the local compressive strain field at the GBs (as revealed by the geometrical phase analysis in the Supplementary Fig. 12) can help accommodating the volume shrinkage during the BCC→FCC transformation, which thus provides the internal mechanical driving force.

In the text, the “volume shrinkage” rather than “volume expansion” was used to refer to the volume change accompanying the BCC→FCC structural change. The density of the IGC sample is about 97% of the as-cast sample, which indicates that the IGC processing is capable to produce a highly dense material. In addition, the

observed structural change and element re-distribution are not relevant to the pores and flaws. Thus, it is suggested that the porosity is not an issue in the diffusional transformation in the present case.

- The authors never discuss about the solid solution hardening, this is quite surprising because they have to admit that the formation of the fcc Ni rich phase should lead to a Ni depletion of the Fe bcc phase and consequently to a reduction of the solid solution hardening.

Response: We agree that the issue of solid solution strengthening of Ni in the BCC iron should be discussed, concerning the consequences of the Ni re-distribution. The substitutional Ni strengthens the lattice of BCC iron^[1]. Therefore, the Ni depletion in the BCC phase due to the formation of a Ni-rich FCC phase should lead to a reduction of shear resistance. However, this softening effect was not observed in the experiments. The present results consistently show the strengthening effect due to the formation of the FCC phase. This can be explained by the scenario that the plastic deformation is controlled by dislocation nucleation at grain boundaries. In that case, the strengthening due to a high dislocation nucleating stress will overwhelm the reduction of strength due to a decrease of the Ni concentration in solid solution.

References

[1] W.C. Leslie, Iron and its dilute substitutional solid solutions, Metallurgical and Materials Transactions B 3, (1972) 5-26.

Changes to the manuscript: In the revised manuscript, we have discussed the solid solution hardening effects of the Ni addition in Fe lattice. The modified part has been highlighted in red.

Page 12, Line 296-302, main text.

Report of reviewer 2-- NCOMMS-22-02265A

Reviewer #2 (Remarks to the Author):

The manuscript by Wu et al. reported a ultra-hardening phenomenon in nanograined metals. By introducing grain segregation and dislocation-free grain structures, nanograined Fe-Ni alloy with d of tens of nanometers were strengthened to a ultrahigh level. Multiscale characterizations have been applied to reveal the microstructural origin of ultrahigh hardening, which was attributed to the activation of dislocation

sources at GBs in the dislocation-free nanograins. Though this work is interesting, major clarifications need be further addressed, especially regarding the origin of observed-phenomenon:

General Response: The authors are grateful to the reviewer for his/her comments and insightful evaluation of this work. We have improved the manuscript, according to the replies to the comments and questions. The detailed responses to the comments are listed below.

1. In introduction, pre-existing dislocations are not always the dominating factor of the plasticity of nanograined metals. It is structure dependent. And the softening is mainly induced by the strain localization rather than grain growth.

Response: The authors agree that the deformation mechanism of nanograined metals is structure dependent. And the particular part of the Introduction should indeed be discussed more carefully. Generally speaking, in the grain size range where the Hall-Petch scaling law works, the plastic deformation of nanograined metals is determined by the dislocation mechanism. Under mechanical loading, dislocation sources, which are either pre-existing lattice dislocations or the GB sources, are activated, and the glide of dislocations provides the carriers for plastic strain^[1-2]. It should be noticed that in nanograined metals, the dislocation storage becomes insignificant^[3-5]. When the grain size is further reduced to the so-called extremely fine nanograined structures with grain size of a few nanometers, the plastic deformation mechanism is changed. It is suggested that GB migration is dominant in the plasticity in that case^[6-7], which contributes to the observed inverse Hall-Petch effect. But when the GBs are stabilized by alloying element segregation or GB relaxation, the activation of dislocation sources at GBs is again dominant in the onset of plasticity in the extremely fine nanograined metals^[6].

The strain-induced softening could be due to the plastic strain localization. But in the Introduction, the mentioned “softening” is the result of the competition between dislocation annihilation and grain growth during annealing but not during mechanical loading. The statement has been improved for better clarity.

References

- [1] X. Huang, N. Hansen, N. Tsuji, Hardening by Annealing and Softening by Deformation in Nanostructured Metals. *Science* 312, (2006) 249-251.
- [2] A. Hasnaoui, H. Van Swygenhoven, P.M. Derlet, On non-equilibrium grain boundaries and their effect on thermal and mechanical behaviour: a molecular dynamics computer simulation, *Acta Materialia* 50, (2002) 3927–3939.
- [3] M. Chen, E. Ma, K.J. Hemker, H. Sheng, Y. Wang, X. Cheng, Deformation twinning in

nanocrystalline aluminum, *Science* 300, (2003) 1275-1277.

[4] Z. Shan, E.A. Stach, J.M.K. Wiezorek, J.A. Knapp, D.M. Follstaedt, S.X. Mao, Grain Boundary-Mediated Plasticity in Nanocrystalline Nickel, *Science* 305, (2004) 654-657.

[5] I.A. Ovid'ko, R.Z. Valiev, Y.T. Zhu, Review on superior strength and enhanced ductility of metallic nanomaterials, *Progress in Materials Science* 94, (2018) 462-549.

[6] J. Hu, Y. N. Shi, X. Sauvage, G. Sha, K. Lu, Grain boundary stability governs hardening and softening in extremely fine nanograined metals. *Science* 355, (2017) 1292-1296.

[7] T. J. Rupert, D. S. Gianola, Y. Gan, K. J. Hemker, Experimental Observations of Stress-Driven Grain Boundary Migration. *Science* 326, (2009) 1686-1690.

Changes to the manuscript: The statement has been improved for better clarity in the revised manuscript. The modified part has been highlighted in red.

Page 2, Line 32-35 and 37-46, main text.

2. How was the grain size measured? It is necessary to provide the grain size distributions for Fig. 1a-b. Also, what is the grain size of UFG samples? More detailed TEM characterization needs to be provided to show the structure of different samples.

Response: The grain size was measured from the bright-field transmission electron microscope (BF-TEM) images, following the ASTM standard E112-13. Since the grains are equiaxed, we employed the Heyn intercept method in the standard way^[1]. Firstly, a line was superimposed over the BF-TEM image. Then the intersections of the line and grain boundaries were marked out, providing the length of each interval. More than 400 grains were measured on each sample. According to the reviewer's suggestion, the grain size distributions, as shown below in Figure R3, were inserted in the Fig. 1a and b in the revised manuscript.

The lamellar thickness of the cold-rolled Fe-Ni sample was calculated by TEM results ($\sim 195 \pm 87$ nm, shown in the Figure R4a and Supplementary Fig. 9a). Concerning the Fe and Ni elemental distribution (Figure R4c and d), no segregation can be observed in the ultra-fine grained (UFG) Fe-Ni alloy.

Figure R3. The grain size distribution of (a) the as-IGC and (b) annealed (300 °C for 1h) nanograined Fe-Ni samples.

Figure R4. BF TEM image and elemental distributions of the cold-rolled Fe-Ni sample. (a) BF TEM image of the cold-rolled Fe-Ni sample, the inserted histogram shows the lamellar thickness distribution. (b) HAADF result of the cold-rolled Fe-Ni sample. (c and d) The corresponding STEM-EDX elemental maps of b. No segregation can be observed.

References

[1] Standard test methods for determining average grain size; E112. Annual book of ASTM standards. Philadelphia: American Society for Testing and Materials; 2000. p. 245.

Changes to the manuscript: The modified parts in the revised manuscript have been highlighted in red.

Page 4, Fig. 1 a and b, main text.

Page 5, Line 133-135, main text.

Page 15, Line 392-395, methods.

Page 10, Newly added Supplementary Fig. 9, Supplementary Information.

3. Page 4. I think it is kind of difficult to exclude the GB mechanism based on current results. In Extended Figure 3, some large grains seem formed beneath the indentation surface? It is better to provide some zoomed-in dark field image. And what are the grain structures under the side surfaces of indentation? The deformation can be different with large shear stress.

Response: The GB mechanism was examined by the observation of the deformed microstructure beneath the indents. Zoomed-in dark-field and bright-field TEM images are given below in Figure R5. We carefully measured the grain size distribution of the microstructure beneath the indentation (0-300 nm from the surface, denoted as surface region) and of those relatively far from the indentation (300-600 nm from the surface, denoted as deep region) of different samples, and the statistical results are shown in Figure R6. The average grain sizes of surface region and deep region for the as-IGC sample are 16 nm and 15.8 nm, and those for the annealed sample are 16.7 nm and 15.9 nm, respectively. Thus the average grain size and the grain size distribution are similar for both deformed microstructures, indicating the absence of grain growth induced by deformation. This is in obvious contrast to the case when GB migration dominates the plastic deformation. As an example referred from [1], in the Ni-Mo alloy with an initial grain size of 5 nm, very large grains up to 150 nm are observed close to the indented surface (Figure R7), and obvious grain growth was also detected for the microstructure at depths of ~300 nm (Figure R7(E)).

In addition, the grain shape is generally equiaxed, without parallel GB alignment that indicates the activation of GB sliding as found in literature^[2,3]. The comparison in the regions of different distances to the side surface of indentation thus suggests the same deformation mode.

Besides the absence of grain growth and grain structure change as revealed by the TEM characterization, the hardness of as-IGC samples with different grain sizes fall in line with the Hall-Petch relationship (Figure 1d). This also suggests a dislocation-mediated, rather than GB-dominated, plastic deformation mechanism.

Figure R5. The BF-TEM and DF-TEM images of (a-b) the as-IGC and (c-d) annealed (300 °C for 1h) Fe-Ni nanograins underneath the indented surface.

Figure R6. Grain size distribution of the as-IGC Fe-Ni nanograins under the indented surface (a) 0-300 nm and (b) 300-600 nm. Grain size distribution of the annealed (300 °C for 1h) Fe-Ni nanograins under the indented surface (a) 0-300 nm and (b) 300-600 nm.

Figure R7. (A) A typical bright-field TEM image taken underneath the indented surface (indicated by the dashed line) in the as-deposited Ni–14.2% Mo sample after microhardness test, with (B to D) corresponding darkfield images at different depths from the surface (indicated by arrows). (E) The cumulative fraction of grain area versus grain size in the subsurface layer of the as-deposited and the as-annealed Ni –14.2% Mo sample before and after micro-hardness test, respectively. For the as deposited sample, two sets of data at depths of ~300 and ~ 600 nm from the indented surface are included. ^[1]

References

- [1] J. Hu, Y.N. Shi, X. Sauvage, G. Sha, K. Lu, Grain boundary stability governs hardening and softening in extremely fine nanograined metals, *Science* 355, (2017) 1292-1296.
- [2] N.A. Mara, A.V. Sergueeva, T.D. Mara, S.X. McFadden, A.K. Mukherjee, Superplasticity and cooperative grain boundary sliding in nanocrystalline Ni3Al, *Materials Science and Engineering A* 463, (2007) 238–244.
- [3] Y. Ivanisenko, L. Kurmanaeva, J. Weissmueller, K. Yang, J. Markmann, H. Rösner, T. Scherer, H.J. Fecht, Deformation mechanisms in nanocrystalline palladium at large strains, *Acta Materialia* 57, (2009) 3391-3401.

Changes to the manuscript: In the revised manuscript, we have provided more detail results and explanation to preclude the GB mechanism. The modified parts have been highlighted in red.

Page 4, Line 106-107, main text.

Page 10, Line 253-260, main text.

Page 4, Newly added Supplementary Fig. 3, Supplementary Information.

Page 5, Newly added Supplementary Fig. 4, Supplementary Information.

4. Fig. 2 cannot represent the typical structural feature of the annealed structures. The GBs with Ni segregations should play more important roles on the hardening, rather than the cluster with high Ni concentration. Similar behaviors have been widely reported in literatures.

Response: Fig. 2 is showing the features of element distribution, while the structural features are characterized by the TEM and XRD results as shown in Fig. 1, 3 and 4. We agree that the GBs segregation could be important in the annealing-induced hardening in nanograined alloys, which can be found in the earlier reports^[1-3]. However, generally speaking, the increase in hardness caused by GB segregation is less than ~20%, except for the case of the extremely-fine nanograined Ni-Mo alloy dominated by GB-mediated plastic mechanism, which obtains a huge strengthening (the magnitude of 6.3 GPa, hardened by 125%) due to the enhanced grain boundary stability^[4].

However, in the present study, the GBs with Ni segregation are not suggested as a significant contribution to the ultra-hardening phenomenon. The Ni segregation is already significant in the as-IGC sample, while the hardness of this sample is in line with the Hall-Petch plot of iron alloys (Fig. 1d). But the Ni re-distribution and further enrichment to the GBs after annealing do have other mechanical consequences. The Ni segregation is essential in triggering the diffusional phase transformation that wipes out the existing lattice dislocations. The resulting “dislocation-free” structure necessitates the activation of dislocation sources at GBs, leading to the ultra-hardening effect.

References

- [1] R.Z. Valiev, N.A. Enikeev, M.Y. Murashkin, V.U. Kazykhanov, X. Sauvage, On the origin of the extremely high strength of ultrafine-grained Al alloys produced by severe plastic deformation, *Scripta Materialia* 63, (2010) 949-952.
- [2] T. Shen, R. Schwarz, S. Feng, J. Swadener, J. Huang, M. Tang, J. Zhang, S. Vogel, Y. Zhao, Effect of solute segregation on the strength of nanocrystalline alloys: Inverse Hall–Petch relation, *Acta Materialia* 55, (2007) 5007-5013.
- [3] Y.M. Wang, S. Cheng, Q.M. Wei, E. Ma, T.G. Nieh, A. Hamza, Effects of annealing and impurities on tensile properties of electrodeposited nanocrystalline Ni, *Scripta Materialia* 51, (2004) 1023-1028.
- [4] J. Hu, Y.N. Shi, X. Sauvage, G. Sha, K. Lu, Grain boundary stability governs hardening and softening in extremely fine nanograined metals, *Science* 355, (2017) 1292-1296.

Changes to the manuscript: In the revised manuscript, we have given more detail description to clarify the influence of the Ni segregation at GBs in our case. The modified parts have been highlighted in red.

Page 6, Line 164-165, main text.

Page 7, Line 166, main text.

Page 13, Line 329-334, main text.

5. Are there any contribution of impurities to the hardening? How to exclude its contribution? The impurity content of 0.7 at% is not so low, and thus needs to be discussed.

Response: In general, the existence of impurities is an issue for clarification when concerning the stability and mechanical properties of nanograined metals. But we believe that the contribution of impurities is not significant in the present case. As mentioned on Page 5, Line 126-131 (main text) and shown in Figure R1, the content of impurities (C, O, N) and their distributions was found to keep unchanged after annealing at 300 °C for 1 h. Compared with the significant re-distribution of Ni atoms during annealing (as shown in Fig. 2), the contribution of the impurities to the annealing-induced ultra-hardening phenomenon is not suggested to be significant.

Changes to the manuscript: The modified parts in the revised manuscript have been highlighted in red.

Page 5, Line 130-131, main text.

Page 9, Newly added Supplementary Fig.8, Supplementary Information.

6. The so-called Phase transformation may not be appreciated to describe the structural transformation during annealing. It is a process similar to the precipitation or GB aggregation of Ni-enriched clusters. And also, the contribution of so-called phase transformation to hardening should be low. The hardening should be mainly originated from the reduced GB mobility due to the GB pinning, as well as the change of defect nucleation stress at aggregated GB, which has been well-studied in literatures.

Response: During the annealing treatment, the concurrence of the BCC-FCC structural change and the diffusion process of Ni re-distribution was observed. The formation of a new phase with a different composition to the matrix could thus be properly described by the term of diffusional transformation. The diffusional nature of the structural change and the small size of the newly-formed Ni-rich FCC phase make it similar to a precipitation process. But the precipitation reaction is actually also a kind of diffusional transformation^[1]. The present BCC-FCC transformation also differs from the GB aggregation of Ni-enriched clusters because of the well-defined structural change.

In the present interpretation, the diffusional transformation results in the exhaustion of lattice dislocations due to the formation of the Ni-enriched FCC phase at GBs. A correlation between the increase of hardness and the reduction of dislocation density has been substantiated by the systematic experimental results. Therefore, the

occurrence of a phase transformation is a pathway to achieve a ‘dislocation-free’ structure for the ultra-hardening in nanograined metals. In this sense, the previous title of the paper “*Ultra-hardening of nanograined metals by diffusional transformations*” is not precisely describing the major findings. We have changed the title to “*Dislocation exhaustion and ultra-hardening of nanograined metals by diffusional transformations*”, in order to demonstrate the underlying physics more clearly.

The effect of reduced GB mobility due to the GB pinning is not suggested to be responsible to the ultra-hardening phenomenon in this case, because stress-driven grain growth was not observed in the present nanograined Fe-Ni alloy either in the as-IGC or the annealed state. But the change of defect nucleation stress at GBs by annealing is an interesting point for discussion. When the pre-existing lattice dislocations are fully annihilated by annealing, the onset of plasticity becomes dominated by the GB dislocation emission processes. The dislocation emission is thus controlled by the GB structure, which could be tuned by heat treatment^[2-4] and is believed to contribute to the continuous hardening during the prolonged annealing at 300 °C as shown in Fig. 1c.

References

- [1] D. A. Porter, K. E. Easterling, Phase transformations in metals and alloys, Van Nostrand Reinhold, 1992.
- [2] X. Huang, N. Hansen, N. Tsuji, Hardening by Annealing and Softening by Deformation in Nanostructured Metals. Science 312, (2006) 249-251.
- [3] E. Ma, T. D. Shen, X. L. Wu, Less is more. Nature materials 5, (2006) 515-516.
- [4] A. Hasnaoui, H. Van Swygenhoven, P.M. Derlet, On non-equilibrium grain boundaries and their effect on thermal and mechanical behaviour: a molecular dynamics computer simulation, Acta Materialia 50, (2002) 3927–3939.

Changes to the manuscript: According to the reviewer’ comments, we have improved the discussion on the diffusional transformation and its hardening effects. The modified parts in the revised manuscript have been highlighted **in red**.

Page 1, Line 1, main text.

Page 6, Line 155-158, main text.

Page 13, Line 329-340, main text.

7. What is the diffusional transformation really mean? A clear definition/description with more evidences need to be provided. Complex GB processes and element redistribution should be involved in this process. The lack of grain growth also needs to be discussed along with this diffusional transformation.

Response: The authors appreciate the reviewer for the questions on the definition of diffusional transformation, which could also be referred to the response to the previous question (NO.6). When annealing the nanograined Fe-Ni alloy, the concurrence of the BCC-FCC structural change and the Ni re-distribution to form a Ni-rich FCC phase was clearly characterized by a combination of synchrotron in-situ XRD experiment, TEM observations and atom probe tomography. Therefore, the formation of a new phase with a different composition to the matrix should be properly described by the term of diffusional transformation. We agree with the comments of the referee that the abovementioned diffusional transformation, including structural change and element redistribution, and the resulting process of lattice dislocation annihilation occur at the grain boundary regions.

The grain size of the as-IGC nanograined Fe-Ni alloy (15 nm) was maintained during the isothermal annealing at 300 °C for 1 h. Annealing at a higher temperature (e.g. 500 °C as shown in Supplementary Fig. 2) results in an increase of the grain size (90 nm). The thermal stability of the present nanograined Fe-Ni alloy is higher than the nanograined pure iron prepared also by the IGC technique^[1]. This is probably attributed to the Ni segregation and the formation of the FCC phase at the GBs during the low-temperature annealing, which pin the GB migration and retard grain growth. The formation of the Ni-enriched FCC phase is facilitated by the enhanced diffusivity in the nanograined structure, and also by the initial intergranular Ni enrichment that provides the chemical driving force for the transformation. These favorable thermodynamic and kinetic conditions are important for a more efficient formation of pinning particles.

References

[1] J. C. Holzer, R. Birringer, J. Eckert, C. E. Krill III & W. L. Johnson, Relaxation and Grain Growth Behavior of Nanocrystalline Iron, MRS Online Proceedings Library 272, (1992) 283–288.

Changes to the manuscript: The modified parts in the revised manuscript have been highlighted **in red**.

Page 6, Line 155-158, main text.

Page 11, Line 290-292, main text.

Page 12, Line 293-296, main text.

Page 13, Line 329-340, main text.

8. The hardening reported in this paper was simply attributed to dislocation-free-induced GB nucleation. Based on my knowledge, several other factors should contribute to the high hardness reported in this paper, including low dislocation density, GB segregation and the resultant reduced GB mobility, as well as the

segregation-induced change of dislocation nucleation stress from GB. The change of GB dynamics should play more important roles.

Response: The strengthening mechanism in the ultra-hardening of the nanograined Fe-Ni alloy was explored by a systematic investigation of the structural and chemical changes during the annealing treatment and by a careful analysis of the possible micromechanisms.

The most significant annealing-induced hardening by GB segregation and GB relaxation has been reported for the so-called extremely fine nanograined metals with grain sizes of a few nanometers^[1]. In these cases, the grain size is so small that the nanograins are presumably dislocation free. Therefore, the characteristics of GBs determines the onset of plasticity and thus the strength. It is suggested that the GB migration becomes the dominant plastic deformation mechanism and leads to the inverse Hall-Petch effect. But when the GBs are stabilized by segregation and local relaxation, stress-driven GB migration is inhibited. Thus the plastic deformation of such nanograined polycrystals is again determined by the dislocation nucleation at GBs. In addition, the local segregation or structural relaxation could influence the GB dislocation nucleation stress^[2-4].

The present IGC nanograined Fe-Ni alloy involves a grain size of 15 nm. Firstly, it is outside the regime of the inverse Hall-Petch effect, and the absence of stress-driven grain growth is evidenced. The grain boundary migration is not a significant deformation mechanism according to the experimental observations. Secondly, the as-IGC nanograined Fe-Ni alloy is initially decorated by intergranular Ni enrichment, while the hardness is still comparable to iron alloys with similar grain sizes (Fig. 1d). This thus is not suggesting the hardening due to GB segregation of Ni. However, it is possible that the GB structure could be changed by the further Ni segregation and the BCC-FCC structural change during annealing, which might influence the dislocation nucleation stress at GBs. This could be relevant to the continuous hardening (up to 10.8 GPa) during the prolong annealing at 300 °C, as shown in Fig. 1c. Therefore, the change of GB dynamics should indeed be taken into account in the interpretation of the annealing-induced ultra-hardening phenomenon.

References

- [1] J. Hu, Y. N. Shi, X. Sauvage, G. Sha, K. Lu, Grain boundary stability governs hardening and softening in extremely fine nanograined metals. *Science* 355, (2017) 1292-1296.
- [2] R.Z. Valiev, N.A. Enikeev, M.Y. Murashkin, V.U. Kazykhanov, X. Sauvage, On the origin of the extremely high strength of ultrafine-grained Al alloys produced by severe plastic deformation, *Scripta Materialia* 63, (2010) 949-952.
- [3] T.J. Rupert, J.R. Trelewicz, C.A. Schuh, Grain boundary relaxation strengthening of nanocrystalline

Ni–W alloys, Journal of Materials Research 27, (2012) 1285-1294.

[4] O. Renk, A. Hohenwarter, K. Eder, K.S. Kormout, J.M. Cairney, R. Pippan, Increasing the strength of nanocrystalline steels by annealing: Is segregation necessary?, Scripta Materialia 95, (2015) 27-30.

Changes to the manuscript: In the revised manuscript, the discussion has been improved based on the above reply, and the modified parts have been highlighted in red.

Page 10, Line 253-260, main text.

Page 13, Line 329-340, main text.

9. “Dislocation-free” cannot represent the main structural feature of the annealed samples, which needs to be changed in both abstract and the main text.

Response: The low-temperature annealing induces several interplaying structural and chemical changes in the nanograined Fe-Ni alloy. The Ni segregation to the grain boundaries and the BCC-FCC structural change at the grain boundaries are the primary microprocesses, which facilitates the significant annihilation of the residual lattice dislocations. When the mechanical properties are concerned, as explained in the replies to the previous questions, the main structural feature of the annealed samples is suggested to be related to the essentially dislocation-free polycrystal microstructure. In that case, there is a lack of pre-existing lattice dislocations for an easy onset of plasticity, and much higher applied stress is required to activate the dislocation sources at grain boundaries, which is contributing to the observed annealing-induced ultra-hardening. Major modifications have been made in the revised version of the paper to better explain the structural and chemical features of the nanograined alloy and the strengthening mechanisms, basing on the replies to the comments and critics and on a deeper discussion relating to the literature.

Changes to the manuscript: In the revised manuscript, we have changed the descriptions in the abstract and the main text accordingly. The modified parts have been highlighted in red.

Page 6, Line 155-158, main text.

Page 11, Line 270-271, main text.

Page 12, Line 299-302, main text.

10. Discussion needs to be further improved in the context of the existing literature in an appropriate manner.

Response: Thanks to the important points raised by the comments and critics from the

referee, the discussion in the manuscript has been improved based on the replies, involving also the relevant important literature.

Report of reviewer 3-- NCOMMS-22-02265A

Reviewer #3 (Remarks to the Author):

The manuscript shows that annealing at moderate temperatures dramatically increases the hardness of an Inert Gas Condensation produced Fe-Ni alloy. The authors argue that this is due to diffusional transformations at the grain boundaries that wipe out the existing dislocations. The resulting dislocation-free structure would hinder the nucleation of follow-up dislocations and account for the strength increase. This theory is interesting, but insufficiently supported by direct evidence.

General Response: The authors appreciate the reviewer for the comments and critics. We agree that the scientific contribution could be much more completed if the interpretation could be supported by direct evidence. But, in fact, it is usually a big challenge for the study on nanograined metals to obtain direct evidence, due to the very small structural scale and the complicated structural features. However, the authors have tried the best to reach a clear conclusion by using a combination of state-of-the-art techniques (e.g. synchrotron XRD, APT, HRTEM). The efforts made in the careful analysis can be seen in the detailed responses below. The manuscript has been improved according to the reviewer's constructive suggestions.

1. Hardness measurements are not suited to measure the incident yield strength (which indeed would be increased by hindering dislocation nucleation), because they introduce a large amount of plastic straining, so the reported hardness corresponds to a flow strength after multiple dislocations have presumably already been emitted and the restriction on dislocation nucleation long overcome.

Response: The authors fully agree with this comment. In principle, the indentation hardness measurement is not the best technique to probe the mechanical response. The indentation is inducing plastic flow beneath the indent and also the elastic constraint of the surrounding material, and the hardness value can be correlated to the flow stress at certain plastic strain^[1]. But this technique is widely used in characterizing the mechanical properties of nanograined materials, due to the limitations in specimen size and in tensile ductility, and still provides a proper ranking of the strength level^[2-4]. In addition, considering the lack of strain hardening of

nanograined metals, the hardness value can be regarded as representing the magnitude of the yield strength.

Probing the incident yield strength of nanograined materials is certainly of great scientific interests. But clear conclusions can only be drawn when several challenges in experimentation and theory are well addressed. Firstly, uniaxial tensile testing is the best technique to measure the mechanical response. But the small dimension of the IGC materials makes it difficult to machine proper tensile specimens. And the high strength level and the existence of minor-but-detrimental flaws usually lead to a brittle fracture behavior. Micropillar compression is a popular alternative, but great attention must be paid on the effect of FIB preparation, the alignment of the testing framework and the precision of strain measurement, which are important details when interpreting the mechanical measurements. Secondly, the correlation between the macroscopic measurement and the micromechanical processes is yet to be established. In the case of nanograined materials, there is inherently an extended elasto-plastic transition^[5]. It means that during the initial stage of plastic deformation, the volume fraction of yielded grains increases gradually, which results in a high apparent strain hardening rate and a significant distribution of internal stresses. A proper interpretation must be based on the combination of high-quality mechanical measurement and a thoughtful micromechanical model.

The authors have improved the discussion. But the point raised by the referee is definitely an excellent topic for future study. The consequences of an annealing-induced diffusional transformation and the effect of dislocation exhaustion should be examined in detail by a combination of carefully-designed experiments and micromechanical modeling accounting for dislocation dynamics.

References

- [1] D. Tabor, The hardness of metals. Clarendon Press, Oxford, 1951.
- [2] J. Hu, Y. N. Shi, X. Sauvage, G. Sha, K. Lu, Grain boundary stability governs hardening and softening in extremely fine nanograined metals. *Science* 355, (2017) 1292-1296.
- [3] X. Ke, J. Ye, Z. Pan, J. Geng, M. F. Besser, D. Qu, A. Caro, J. Marian, R. T. Ott, Y. M. Wang, F. Sansoz, Ideal maximum strengths and defect-induced softening in nanocrystalline-nanotwinned metals, *Nature Materials* 18, (2019) 1207-1214.
- [4] M.A. Meyers, A. Mishra, D.J. Benson, Mechanical properties of nanocrystalline materials, *Progress in Materials Science* 51, (2006) 427-556.
- [5] G. Saada, T. Kruml, Deformation mechanisms of nanograined metallic polycrystals, *Acta Materialia* 59, (2011) 2565-2574.

Changes to the manuscript: The modified parts in the revised manuscript have been highlighted **in red**.

Page 3, Line 87-89, main text.

Page 4, Line 90-91, main text.

2. The second issue is that the authors are reaching their conclusions by ruling out alternative explanations rather than direct evidence, see below. If the dislocation density would be the main factor, why not rationalize the increase in strength by a dislocation model – or at least provide the theoretical strength of Fe-Ni for comparisons to the experimental values?

Response: Due to the multiscale nature of plastic deformation and the complicated structural features in nanograined metals, it is challenging to provide direct evidences (of the elementary micro-processes) for the macroscopic mechanical response and to establish the structure-property relationship. The conclusion in this study was drawn after examining various possible micromechanisms, and the most significant strengthening contribution was identified when being substantiated by the systematic investigation of the structural and chemical changes during the annealing treatment. It would be perfect if direct evidence could be provided, but, as discussed in the previous reply, this requires continuous efforts in addressing the challenges in the development of experimental techniques and modelling methods.

The authors appreciate the suggestion to compare the current results with the theoretical strength, which is helpful in rationalizing the experimental observations. For the BCC iron, the theoretical shear strength, for either the shearing of two neighboring planes^[1] or the homogeneous nucleation of a dislocation^[2], could be estimated as 6.6 GPa (of the order of $G/10$). This corresponds to a Vicker's hardness of ~60 GPa when assuming a random crystal orientation and using the empirical relationship between strength and hardness. This value is several times higher than the observed maximum hardness in this study (10.8 GPa), but they are actually within the same order of magnitude. The discrepancy can be explained by the following reasons: Firstly, the addition of Ni can decrease the elastic modulus of iron^[3] and thus tends to reduce the theoretical shear strength. Secondly, the plastic deformation is presumably controlled by the dislocation nucleation from grain boundaries, which is not in exact agreement with the assumptions for the calculation of the theoretical shear strength. In addition, the local shear stress at the grain boundary regions might prematurely reach the theoretical limit due to the presence of possible stress concentrators, including the incompatible stress due to elastic anisotropy, the ledges in the grain boundaries and the existence of flaws and discontinuities. Therefore, the measured ultrahigh hardness is considered as approaching the magnitude of the theoretical value, when accounting for abovementioned factors.

References

- [1] A. Kelly, Strong Solids. Clarendon Press, 1973.
- [2] T. Zhu, J. Li, Ultra-strength materials, Progress in Materials Science 55, (2010) 710-757.
- [3] W.C. Leslie, Iron and its dilute substitutional solid solutions, Metallurgical and Materials Transactions B 3, (1972) 5-26.

Changes to the manuscript: In the revised manuscript, the discussion has been improved according to the reviewer's comments. The modified parts have been highlighted in red.

Page 12, Line 321, main text.

Page 13, Line 322-329, main text.

3. The conclusions are based on the assumption of a stable microstructure. This critical point is insufficiently supported by data. The authors must provide the full grain size distribution both BEFORE and after heat treatment at 300°C. To me, it appears that the proportion of smaller grains must increase, since <5 nm large fcc grains are nucleated (see Fig. 4b).

Response: In the measurement of grain size distribution, the grains for the FCC and BCC phases were not distinguished. The statistical results are shown in Figure R3 and also provided in Fig. 1a and b, which are similar for the sample before and after annealing at 300 °C for 1 h. The influence of the FCC phase formation on the grain size distribution might probably be limited by the relatively low volume fraction.

It is also worth noticing that, as shown in Fig. 1c, even when a certain degree of grain growth occurs during prolonged annealing, the ultrahigh hardness is still achieved. This indicates that when the plasticity is controlled by the GB dislocation nucleation processes, the ultra-strength can be achieved within a certain range of grain size, associated with a less stringent requirement of grain refinement or the stability of a nanograined microstructure.

Changes to the manuscript: In the revised manuscript, the grain size distributions (before and after heat treatment at 300°C for 1h) have been inserted in the Fig. 1a-b. The modified parts have been highlighted in red.

Page 4, Fig. 1a and b, main text.

Page 15, Line 392-395, methods.

4. My own feeling is that the increase of hardness is rather connected to the nucleation of this fine-dispersed (see Fig. 2c as well as the fine lamellar structure in ExFig 7a) phase and to the juxtaposition of bcc/fcc phases, than from the dislocation-free

condition.

Response: The authors appreciate the point raised by the referee, which definitely deserves further discussion in the manuscript. It would be intuitive to consider the precipitation hardening and the composite strengthening in a two-phase system. But since the small FCC islands are formed at the GBs of the BCC matrix, they are not playing the role of obstacles to the glide of lattice dislocation as in the case of precipitation hardening. In addition, the FCC second phase is not expected to be responsible for such ultra-hardening effect via a composite mechanism. If we estimate the hardness of the second phase according to the rule of mixtures, we will attain a value of 55 GPa, which is too high a value for a metallic phase constituent in a real microstructure. In addition, we prepared a FCC Fe₅₀Ni₅₀ nanograined sample (grain size: 21 ± 5.5 nm) by IGC (see Supplementary Fig. 15) and obtained a hardness of 4.2 GPa, which is lower than the hardness of the BCC Fe₈₄Ni₁₆ sample with similar grain size and is presumably not a reinforcement.

The above analysis also leads us to consider other possible mechanical consequences of the diffusional transformation, which constitute the originalities of this study. Firstly, the FCC phase is preferably formed at the GBs with the attaching lattice dislocations. As shown by the GPA analysis in the Supplementary Fig. 12, the lattice dislocations attached to GBs are inducing the strain energy that could energetically favor the structural change. In addition, such lattice dislocations at the GB regions are also kinetically favoring the diffusional transformation^[1]. Secondly, the diffusional transformation involves a flux of diffusing species, which facilitates the annihilation of dislocations by climbing. These two effects facilitate the formation of a “dislocation-free” nanograined polycrystal structure. Another possible consequence could be related to the nucleation sites of the transformation at the GBs, which are probably the non-equilibrium locations of GBs. Since the non-equilibrium locations at the GBs are also the potent sites for dislocation nucleation^[2]. The formation of the FCC phase is thus increasing the apparent nucleation stress of dislocations, which will be more significant when the pre-existing lattice dislocations are highly exhausted and when the characteristics of the GBs dominate the onset of plasticity. This is indicated by the sustained increase of hardness during the prolonged annealing at 300 °C as shown in Fig. 1c.

References

[1] G. Wilde et al., Plasticity and Grain Boundary Diffusion at Small Grain Sizes, *Advanced Engineering Materials* 12, (2010) 758-764.

[2] I.A. Ovid'ko, A.G. Sheinerman, R.Z. Valiev, Dislocation emission from deformation-distorted grain boundaries in ultrafine-grained materials, *Scripta Materialia* 76, (2014) 45-48.

Changes to the manuscript: According to the reviewer's comments, we have improved the discussion in the revised manuscript. The modified parts have been highlighted in red.

Page 11, Line 272-277, main text.

Further remarks:

5. The authors are not discussing the influence of the residual 3% porosity and 0.7% impurities, which is also a major difference to the cold-rolled sample and might also play a role in the phase transformation.

Response: The authors agree that the influence of the residual 3% porosity and the 0.7 at.% impurities should be considered in the materials prepared by IGC. In the present case, the IGC Fe-Ni nanograined alloy has 97% of the density of the as-cast sample, which means that the IGC system is capable to produce a highly dense material. The observed structural change and element re-distribution are not relevant to the pores and flaws. Therefore, the porosity is not considered as an issue in the diffusional transformation. However, it is figured out that such minor amount of porosity could be detrimental to the tensile properties and leads to a brittle tensile behavior.

The 3D chemical maps of the impurities for the nanograined Fe-Ni sample before and after annealing are provided in the Figure R1 (Supplementary Fig. 8). The impurities (C, O, N) are homogeneously distributed within the sample, and the segregation and re-distribution are not observed after annealing at 300 °C for 1 h. Compared with the significant re-distribution of Ni atoms during annealing (as shown in Fig. 2), the unaltered distribution of the impurities is not suggesting a significant role in the phase transformation and the annealing-induced ultra-hardening phenomenon.

The big difference concerning the phase transformation in comparison to the cold-rolled sample deserves further explanation. The cold-rolled microstructure also involves substantial GBs and dislocations that are kinetically favoring the transformation. However, as shown by the additional TEM-EDX results in the Figure R4 (Supplementary Fig. 9), the initial intergranular Ni enrichment is absent in the cold-rolled sample, and thus the diffusional transformation cannot be triggered at a lower temperature due to the insufficient chemical driving force.

Changes to the manuscript: In the revised manuscript, the discussions have been improved based on the above reply. The modified parts have been highlighted in red.

Page 5, Line 130-131 and 133-135, main text.

Page 9, Newly added Supplementary Fig.8, Supplementary Information.

Page 10, Newly added Supplementary Fig.9, Supplementary Information.

6. It is not clear to me how the authors can experimentally quantify and distinguish the grain boundary dislocations from the grain-interior dislocations.

Response: Firstly, the authors would like to clarify the concept of grain boundary dislocations (GBDs). The concept of GBDs was developed to describe the structure of grain boundaries based on the geometric descriptions of two interpenetrating lattices, including the coincidence site lattice (CSL-lattice), Σ -lattice, and displacement shift complete lattice (DSC-lattice). GBDs are line defects lying in a grain boundary to accommodate the misorientation/deviation between the two crystals adjoining the boundary. GBDs are thus part of the equilibrium structure of boundaries^[1-4]. The Burgers vectors of the GBDs depend on the periodic structure of the grain boundary, and are not crystal lattice vectors. A representative schematic diagram of a GBD and its Burgers vector in a DSC-lattice is provided in Figure R8^[5]. It is a $(3\bar{1}0)\Sigma = 5$ boundary in the FCC structure, O and V symbols represent ... ABA... stacking of (002) planes, blue line denotes the Burgers circuit according to which the Burgers vector of GBD is $1/10[\bar{1}\bar{3}0]$, i.e., a small magnitude. Therefore, the scale of GBDs is too fine to be resolved readily, and they have no long-range stress field^[2].

In our manuscript, we are dealing with lattice dislocations but not GBDs. In the text, the terms of “**dislocations at the GB region**”, “**dislocations attached to GBs**”, and “**dislocations adjacent to GBs**” were used in order to avoid the confusion with GBDs. Note that the dislocations adjacent to grain boundaries and the grain-interior dislocations are both lattice dislocations, which can be readily distinguished from GBDs by identifying their Burgers vectors in HRTEM images. As a matter of fact, we indeed conducted the Burgers circuits to analyze the dislocations adjacent to grain boundaries (Fig. 4a, Supplementary Fig. 12 and 14), showing that their Burgers vectors are typical crystal lattice vectors which are fundamentally different with the GBDs.

The lattice dislocation density of the nanograined Fe-Ni alloys was quantified by using the modified Williamson–Hall method to analyze the data of the synchrotron XRD experiments (see Methods for details). This method is based on two models: (1) the phenomenological model accounting for the elastic anisotropy of the crystals; (2) the dislocation model on the basis of the mean square strain of dislocated crystals^[6-9]. The order-dependent microstrain broadening in XRD profiles stems from the lattice dislocations which produce long-range stresses. The GBDs, which is not involving a long-range stress field, are not contributing to the order-dependent broadening in the XRD profiles. The effect of lattice dislocations in the GB region on the strain

broadening in XRD of nanograined materials has recently been verified in Ref^[6] by both detailed characterization and MD simulation.

Figure R8. Schematic diagram of a GBD in $(310) \Sigma = 5$ boundary in FCC structure^[5].

References

- [1] W. Bollmann, On the geometry of grain and phase boundaries: I. General theory, *Philosophical Magazine* 16, (1967) 363-381.
- [2] J.P. Hirth, R.W. Balluffi, On grain boundary dislocations and ledges, *Acta Metallurgica* 21, (1973) 929-942.
- [3] R.C. Pond, D.A. Smith, On the absorption of dislocations by grain boundaries, *Philosophical Magazine* 36, (1977) 353-366.
- [4] D.A. Smith, C.M.F. Rae, Development in annealing and structure and mobility of grain boundaries in pure metals, *Metal Science* 13, (1979) 101-107.
- [5] D.A. Smith, R.C. Pond, Bollmann's 0-lattice theory; a geometrical approach to interface structure, *International Metals Reviews* 21, (1976) 61-74.
- [6] Z. Zhang, É. Ódor, D. Farkas, B. Jóni, G. Ribárik, G. Tichy, S.-H. Nandam, J. Ivanisenko, M. Preuss, T. Ungár, Dislocations in Grain Boundary Regions: The Origin of Heterogeneous Microstrains in Nanocrystalline Materials, *Metallurgical and Materials Transactions A* 51, (2019) 513-530.
- [7] T. Ungár, A. Borbély, The effect of dislocation contrast on x-ray line broadening: A new approach to line profile analysis, *Applied Physics Letters* 69, (1996) 3173-3175.
- [8] T. Ungár, J. Gubicza, G. Ribárik, A. Borbély, Crystallite size distribution and dislocation structure determined by diffraction profile analysis principles and practical application to cubic and hexagonal crystals, *Journal of Applied Crystallography* 34, (2001) 298-310.
- [9] T. Ungár, I. Dragomir, Á. Révész, A. Borbély, The contrast factors of dislocations in cubic crystals: the dislocation model of strain anisotropy in practice, *Journal of Applied Crystallography* 32, (1999) 992-1002.

7. The references are split in two by the methods section, which makes them very inconvenient to read.

Response: We have combined the split parts of the references in the revised manuscript for the convenience of reading.

REVIEWER COMMENTS

Reviewer #1 (Remarks to the Author):

The authors have provided a clear and detailed answer to each points raised by the reviewers. They have modified the manuscript taking into account most of the comments and increasing significantly its quality. In this revised form, the paper seems to me fully appropriate for publication in Nature Communications.

Xavier Sauvage

Reviewer #2 (Remarks to the Author):

"Diffusional transformation" cannot well summarize the findings of this paper, and may cause some confusion. During thermal treatment, all microstructural transformations of materials are more or less diffusional-related. Besides, "diffusional transformation" also does not represent the main structural feature or design of nanograined metals of this paper. Thus, the title and main text are suggested to further improve.

Reviewer #3 (Remarks to the Author):

Thank you to the authors for providing a detailed answer to all three reviews. On many topics, they have significantly improved their manuscript.

However, it appears to me that a core concern of reviewer #2 and myself has not been fully addressed yet: as it stands, the manuscript still presents the "dislocation-free" condition of the sample as the main and original feature accounting for the high strength of the material in categorical terms. As already pointed out by reviewer #2 and myself, there is no sufficient evidence that this effect (low dislocation density and dislocation emission at the grain boundaries) would be dominant over e.g. the segregation of elements at grain boundaries, with consequences on grain boundary sliding, phase duality and precipitation strengthening.

While the report of an unconventionally high strength would be of great interest to the readership of Nature Communications, I believe that the underlying theory is presented in too bold a manner – since it is lacking direct evidence. Cautious wording would be warranted, especially in the abstract and conclusions.

Report of reviewers-- NCOMMS-22-02265A

Reviewer #1 (Remarks to the Author):

The authors have provided a clear and detailed answer to each points raised by the reviewers. They have modified the manuscript taking into account most of the comments and increasing significantly its quality. In this revised form, the paper seems to me fully appropriate for publication in Nature Communications.

Xavier Sauvage

Response: The authors are very grateful to the positive evaluation of the reviewer on the revised manuscript.

Reviewer #2 (Remarks to the Author):

"Diffusional transformation" cannot well summarize the findings of this paper, and may cause some confusion. During thermal treatment, all microstructural transformations of materials are more or less diffusional-related. Besides, "diffusional transformation" also does not represent the main structural feature or design of nanograined metals of this paper. Thus, the title and main text are suggested to further improve.

Response: The authors appreciate that the reviewer agrees with most of the clarifications and modifications of the revised manuscript.

We agree with the comment about the use of 'diffusional transformation' in the title and main text. Indeed, merely the diffusional nature cannot capture the whole feature of the ultra-hardening of the nanograined alloys. Compared to the strengthening due to the dislocation obstacles introduced by phase transformation (e.g. precipitation hardening), the present study demonstrates that, in the nanograined metals, diffusional transformation at grain boundaries can be explored to engineer the source of mobile dislocations and to lead to a shift of controlling mechanism of plastic deformation. In brief, the grain-boundary mediated transformation provides a pathway to achieve a 'dislocation-exhausted' condition for the ultra-hardening in nanograined metals. Therefore, the title of the paper has been changed to "*Dislocation exhaustion and ultra-hardening of nanograined metals by grain-boundary mediated transformations*", and the relevant parts in the abstract and the main text have been changed accordingly and highlighted in red.

Reviewer #3 (Remarks to the Author):

Thank you to the authors for providing a detailed answer to all three reviews. On many topics, they have significantly improved their manuscript.

However, it appears to me that a core concern of reviewer #2 and myself has not been fully addressed yet: as it stands, the manuscript still presents the "dislocation-free" condition of the sample as the main and original feature accounting for the high strength of the material in categorical terms. As already pointed out by reviewer #2 and myself, there is no sufficient evidence that this effect (ow

dislocation density and dislocation emission at the grain boundaries) would be dominant over e.g. the segregation of elements at grain boundaries, with consequences on grain boundary sliding, phase duality and precipitation strengthening.

While the report of an unconventionally high strength would be of great interest to the readership of Nature Communications, I believe that the underlying theory is presented in too bold a manner – since it is lacking direct evidence. Cautious wording would be warranted, especially in the abstract and conclusions.

Response: The authors appreciate the reviewer’s evaluation on the modifications and on the value of publishing this paper.

The authors understand well the concerns of the reviewer. Actually, in the previous revised manuscript and the response letter, we have examined various possible micromechanisms, and suggested that the most significant contribution to the annealing-induced ultra-hardening is the very low dislocation density and dislocation emission at GBs. For example, 1) The GB mechanism was examined by the TEM observations of the deformed microstructure beneath the indents, and neither GB sliding nor grain growth was observed. 2) The precipitation hardening and the composite strengthening in a dual-phase system were also discussed. Since the small FCC islands are formed at the GBs, they are not playing the role of obstacles to the glide of lattice dislocations as in the case of precipitation hardening. And we have provided the experimental results and order-of-magnitude estimation to demonstrate that the composite strengthening mechanism could not be the major contribution. 3) The GBs with Ni segregation are not suggested as a primary and direct contribution to the ultra-hardening phenomenon, because the Ni segregation is already significant in the as-IGC sample, while the hardness of this sample is in line with the Hall-Petch plot of iron alloys (Fig. 1d). However, the further Ni segregation at GBs during annealing might have other possible consequences, which is explained in details below. As shown in the review comment above, the reviewer #2 seems to have been convinced by the clarifications and modifications.

We fully agree that more cautious wording should be warranted in the manuscript and we have modified the text in several instances accordingly. Firstly, the term of ‘dislocation-free’ is indeed categorical and could lose the accuracy. We have changed it to the more appropriate ‘**dislocation-exhausted**’ term to describe the condition of very low dislocation density. Secondly, the structure-property relationship of nanograined metals is a complicated topic, which involves the proper description of the microstructure and the analysis on the competition and interaction of various micromechanisms. The main messages of this work include: 1) pointing out that the pre-existing lattice dislocations in nanograins are important carriers of plasticity; 2) introducing a particular material design strategy, demonstrating the ultra-hardening effect due to dislocation exhaustion and to a shift to the dominant plastic deformation mechanism that is controlled by GB dislocation nucleation. But, as also highlighted in the previous round of revision, when the lattice dislocations are highly exhausted and there is a lack of mobile dislocations, there should also be other possible considerations for the strengthening contributions. One possibility is the GB structure modification by alloying element segregation and phase transformation, which could strongly influence the critical stress for dislocation emission from GBs and thus the onset of plasticity. The new modifications in the abstract and main text have been highlighted in red.

REVIEWERS' COMMENTS

Reviewer #3 (Remarks to the Author):

Thank you to the authors for their comments. When I wrote about the need for a caution wording, I was mostly referring to very definite statements that are insufficiently supported by evidence. For instance, the abstract reads “we show [...] could be strengthened [...] by exhausting the lattice dislocations”. The opinion of this reviewer was and still is that the causality is not fully demonstrated. Rather, this is a (granted, very likely) suggestion by the authors, and should in any case be worded as such (e.g. which we attribute to...). Regarding the new title “grain-boundary mediated transformations”, I could not fully understand what you mean with “mediated”. There is ample evidence in your manuscript that the phase transformation takes place near grain boundaries, but I would not be able to tell if it is them – or rather the consumed dislocations – that are actually feeding this phase transformation.

Report of reviewers-- NCOMMS-22-02265B

Reviewer #3 (Remarks to the Author):

Thank you to the authors for their comments. When I wrote about the need for a caution wording, I was mostly referring to very definite statements that are insufficiently supported by evidence. For instance, the abstract reads “we show [...] could be strengthened [...] by exhausting the lattice dislocations”. The opinion of this reviewer was and still is that the causality is not fully demonstrated. Rather, this is a (granted, very likely) suggestion by the authors, and should in any case be worded as such (e.g. which we attribute to...). Regarding the new title “grain-boundary mediated transformations”, I could not fully understand what you mean with “mediated”. There is ample evidence in your manuscript that the phase transformation takes place near grain boundaries, but I would not be able to tell if it is them – or rather the consumed dislocations – that are actually feeding this phase transformation.

Response: The authors thank the reviewer for the more detailed suggestion to improve the wording of the manuscript. And we have further improved the manuscript in a more cautious way. For instance, the sentence in abstract “we show [...] could be strengthened [...] by exhausting the lattice dislocations” has been changed to “it is shown [...] could be strengthened [...], which we attribute to the dislocation exhaustion”. On Page 9, Line 7, the sentence “The lattice dislocations in the present nanograins are [...]” has been changed to “The above measurements suggest that the lattice dislocations in the present nanograins are [...]”, etc. All the modifications in the text are highlighted in red.

The title is modified, mainly following the suggestion of reviewer #2, which emphasizes that the term of diffusional transformation cannot represent the most significant feature or design of the present nanograined alloy. The authors agree with that and now use the term “grain-boundary mediated transformations” in the title and the text. According to the observations in the main text (Figure 2 and 4) and the supplementary materials (Figure S10), the BCC-FCC transformation during the low-temperature annealing occurs along the grain boundaries (GBs) and particularly at the triple junctions. As is discussed in the manuscript, the dislocations attaching to the GBs, which are consumed during annealing, are kinetically and thermodynamically favoring the transformation. However, a key point of our present study is that the structural change is facilitated by the GB physico-chemical engineering via intergranular Ni segregation. The GBs and dislocations are not necessary condition for the structural transformation at such a low temperature, which is substantiated by the observation that the transformation occurs at a much higher temperature in the cold-rolled UFG counterpart that does not involve the intergranular Ni segregation. Therefore, the term “grain-boundary mediated transformations” is used to demonstrate the location of structural change, to include the diffusional nature and also to indicate the GB physico-chemical engineering strategy which is explained in the abstract and in the text.